# Fibroblast growth factor 21 increases insulin sensitivity through specific expansion of subcutaneous fat

Huating Li[1,2,3], Guangyu Wu[1,4], Qichen Fang[1], Mingliang Zhang[1], Xiaoyan Hui[2,3], Bin Sheng[5], Liang Wu[1,4], Yuqian Bao[1], Peng Li[6], Aimin Xu[2,3] & Weiping Jia[1]

Although the pharmacological effects of fibroblast growth factor 21 (FGF21) are well-documented, uncertainty about its role in regulating excessive energy intake remains. Here, we show that FGF21 improves systemic insulin sensitivity by promoting the healthy expansion of subcutaneous adipose tissue (SAT). Serum FGF21 levels positively correlate with the SAT area in insulin-sensitive obese individuals. *FGF21* knockout mice (FGF21KO) show less SAT mass and are more insulin-resistant when fed a high-fat diet. Replenishment of recombinant FGF21 to a level equivalent to that in obesity restores SAT mass and reverses insulin resistance in FGF21KO, but not in adipose-specific βklotho knockout mice. Moreover, transplantation of SAT from wild-type to FGF21KO mice improves insulin sensitivity in the recipients. Mechanistically, circulating FGF21 upregulates adiponectin in SAT, accompanied by an increase of M2 macrophage polarization. We propose that elevated levels of endogenous FGF21 in obesity serve as a defense mechanism to protect against systemic insulin resistance.

[1] Department of Endocrinology and Metabolism, Shanghai Key Laboratory of Diabetes Mellitus, Shanghai Clinical Center of Diabetes, Shanghai Jiao Tong University Affiliated Sixth People's Hospital, Shanghai 200233, China. [2] State Key Laboratory of Pharmaceutical Biotechnology, Hong Kong, China. [3] Department of Medicine, The University of Hong Kong, Hong Kong, China. [4] Department of Medicine, Shanghai Jiao Tong University School of Medicine, Shanghai 200025, China. [5] Department of Computer Science and Engineering, Shanghai Jiao Tong University, Shanghai 200240, China. [6] Tsinghua-Peking Center for Life Sciences, School of Life Sciences, Tsinghua University, Beijing 100084, China. Huating Li and Guangyu Wu contributed equally to this work. Correspondence and requests for materials should be addressed to A.X. (email: amxu@hku.hk) or to W.J. (email: wpjia@sjtu.edu.cn)

Fibroblast growth factor 21 (FGF21) has recently attracted great attention due to its multiple therapeutic benefits against obesity-related medical complications[1]. Transgenic mice overexpressing FGF21 exhibit resistance to the development of high-fat diet (HFD)-induced obesity[2]. Subsequent studies have demonstrated that injection of FGF21 can lead to a dramatic decline in fasting glucose, insulin, glucagon and triglycerides in obese diabetic rodents[3,4] and rhesus monkeys[5,6]. Long-term administration of FGF21 analogs can improve dyslipidemia and decrease body weight in patients with obesity and type 2 diabetes (T2DM). In addition, a prominent reduction in fasting insulin levels and a robust increase of adiponectin levels are observed in these patients[7,8].

Although the pharmacological effects of FGF21 are widely recognized, the pathophysiological role of FGF21 is still a matter of debate. Endogenous FGF21 acts as a stress-responsive hormone to defend against different metabolic or environmental stress in diverse conditions[9]. Circulating FGF21 was induced by prolonged fasting, along with increased serum transaminases in humans, suggesting that FGF21 may regulate the utilization of fuel derived from tissue breakdown[10]. In addition, FGF21 is required for adaptations to cold exposure by increasing the level of uncoupling protein-1 (UCP-1) and other thermogenic genes in fat tissues[11]. FGF21 also functions to protect against cardiac hypertrophy by repressing inflammation and prompting fatty acid oxidation[12]. Hepatic FGF21 was increased in acetaminophen-induced liver toxicity, thus decreasing oxidative stress and enhancing antioxidant capacity in the liver[13].

Despite its multiple benefits, FGF21 is paradoxically elevated in obesity and diabetes in both animals and humans[14–16]. However, the pathophysiological role of elevated circulating FGF21 in obesity has never been explored. A previous study in animal suggests the existence of FGF21 resistance[17], whereas there is also a study which does not support the existence of FGF21 resistance[18]. To further address the pathophysiological role of elevated FGF21 in obesity, we raised FGF21 concentration in FGF21 knockout (FGF21KO) mice to a level equivalent to those occurring in diet-induced obesity and then monitored the metabolic changes under this condition. Unexpectedly, we found elevated endogenous FGF21 in obesity serves as a defense mechanism against systemic insulin resistance. Furthermore, we uncover specific expansion of subcutaneous fat, but not visceral fat, as a novel mechanism by which FGF21 promotes systemic insulin sensitivity.

## Results

### Association of serum FGF21 with the subcutaneous fat area (SFA).
Central obesity, featured by the expansion of visceral fat, is correlated with insulin resistance, high risk of type 2 diabetes (T2DM), and cardiovascular diseases[19–23]. In contrast, peripheral adiposity, characterized by expansion of subcutaneous fat, is associated with improved insulin sensitivity and a lower risk of developing T2DM and atherosclerosis in comparison to central obesity[24,25]. These individuals can be referred to as insulin-sensitive overweight and obesity (ISO), which often occurs in the early phase of weight gain[26]. The study of FGF21 in individuals with ISO and insulin-resistant overweight and obesity (IRO) can help us to understand whether FGF21 is involved in the protection from the progression to insulin resistance in the development of obesity.

We recruited young and gender-matched individuals with normal weight (NW) (body mass index [BMI] 18.5-24.9 kg m$^{-2}$) ($n = 30$; age, $32.10 \pm 2.52$ years) and overweight or obese (BMI $\geq 25$ kg m$^{-2}$) ($n = 60$; age, $33.02 \pm 4.94$ years). In these individuals with overweight or obesity, ISO is defined as

homeostasis model assessment (HOMA)-IR < 2.5 and IRO is defined as HOMA-IR $\geq 2.5$ ($n = 30$ in each group)[26]. It was worth mentioning

that in individuals we recruited, the BMI and total fat mass were similar between ISO (BMI, $27.53 \pm 1.95$ kg m$^{-2}$; total fat mass, $23.69 \pm 6.20$ kg) and IRO (BMI, $28.18 \pm 2.14$ kg m$^{-2}$; total fat mass, $24.58 \pm 6.88$ kg). Hepatic triglyceride contents were measured by magnetic resonance spectroscopy. As FGF21 levels are closely associated with the degree of hepatic steatosis via PPARα-dependent regulation[27], any individuals with abnormal hepatic triglyceride content (>5.56%) were excluded[28]. All these 90 individuals received abdominal MRI and hyperinsulinemic–euglycemic clamp to determine their fat distribution and insulin sensitivity accurately.

Anthropometric parameters and biochemical indexes were described in Supplementary Table 1. Although the total fat mass was similar in individuals with ISO and IRO, fat distribution differed remarkably in these two groups. SFA in individuals with ISO was significantly higher than that in individuals with IRO. However, visceral fat area (VFA) in individuals with ISO was significantly lower than that in individuals with IRO. SFA to VFA ratio in ISO was significantly higher than that in NW, while the ratio in IRO was lower than that in NW (Fig. 1a–d). The result of hyperinsulinemic–euglycemic clamp confirmed that unlike individuals with IRO, ISO individuals did not have an obvious reduction in glucose infusion rate (GIR) (Fig. 1e), suggesting that the increased fat mass which was mainly displayed in the subcutaneous region led to better insulin sensitivity. Serum adiponectin level significantly decreased in individuals with IRO but remained unchanged in individuals with ISO (Supplementary Table 1). Circulating FGF21 levels in both ISO and IRO groups were higher than those in NW individuals. Notably, we found serum FGF21 levels were markedly higher in individuals with ISO (194.66 pg ml$^{-1}$ [119.66, 256.86]) than individuals with IRO (134.27 pg ml$^{-1}$ [88.95, 207.78]) (Fig. 1f). Furthermore, SFA was positively correlated with serum FGF21 levels in individuals with ISO ($r = 0.450$) (Fig. 1g). However, no significant relationship between VFA and FGF21 was found in these individuals (Fig. 1h). Serum FGF21 was independently associated with SFA after the adjustment for serum adiponectin level. These clinical findings suggested that elevated serum FGF21 in ISO individuals were closely correlated with the increased subcutaneous fat, which may contribute to the maintenance of insulin sensitivity.

To further investigate the role of the co-receptor βklotho in the function of FGF21 on subcutaneous adipose tissue (SAT), we compared βklotho mRNA and protein levels in SAT and visceral adipose tissue (VAT) in individuals with ISO and IRO. It was found that ISO individuals displayed increased βklotho level in SAT. However, this elevation of βklotho in SAT did not exist in individuals with IRO (Fig. 2a, c). On the other hand, βklotho level in the VAT was gradually decreased from ISO to IRO individuals (Fig. 2a, d). No significant difference in the expression of fibroblast growth factor receptor 1 (*FGFR1*) was found in SAT or VAT among NW, ISO and IRO individuals (Fig. 2b). The results in the mice model were consistent with those in humans (Supplementary Fig. 1). These results suggest that upregulation of βklotho in SAT in the early phase of weight gain may translate the increased FGF21 levels into metabolically beneficial functions.

### FGF21 is required for the accumulation of SAT mass.
The positive correlation between FGF21 and subcutaneous fat in humans prompted us to explore this further in mice models. Eight-week-old, male FGF21KO mice and the wild-type (WT) littermates were fed with standard chow (STC) or HFD for 16 weeks. Body weight and composition were measured once

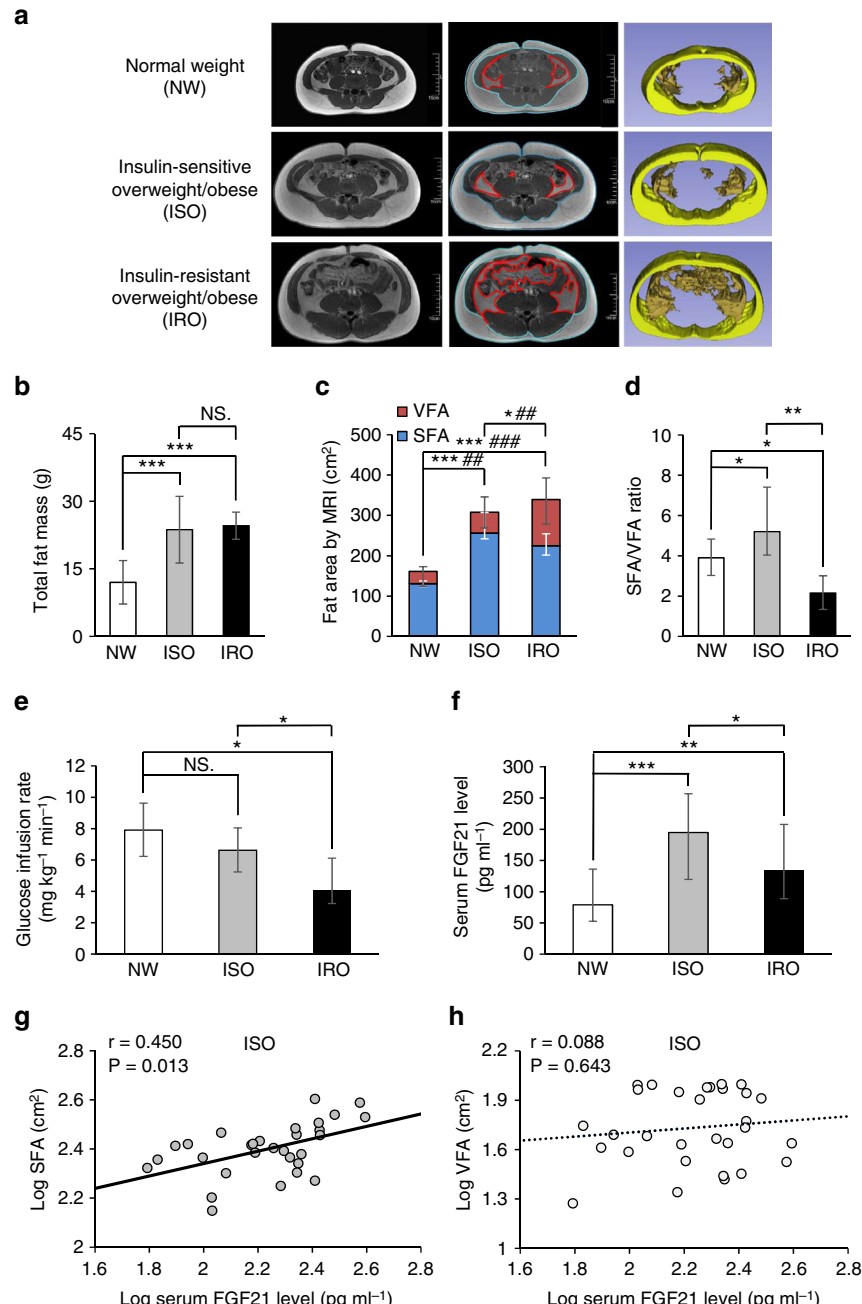

**Fig. 1** Positive association between FGF21 and SFA in ISO individuals. **a** Representative abdominal MRI of three groups of individuals, including NW ($n = 30$), ISO ($n = 30$) and IRO ($n = 30$). Raw (left panel) and marked MRI (middle panel) at navel level, blue lines delineate SFA and red lines delineate VFA, 3D-reconstructed MRI (right panel) after eliminating other organs, yellow represents subcutaneous fat area (SFA) and orange represents visceral fat area (VFA). Scale bar = 10 cm. **b** Total fat mass in three groups. **c** Fat distribution (SFA and VFA contents) evaluated by MRI in three groups. *comparison of SFA, #comparison of VFA. **d** SFA and VFA ratio in three groups. **e** Comparison of glucose infusion rate evaluated by hyperinsulinemic–euglycemic clamp in three groups. **f** Serum FGF21 levels measured by ELISA in three groups. **g,h** Relationship between serum FGF21 levels and **g** SFA as well as **h** VFA in individuals with ISO. Data are presented as mean ± s.d. or median (interquartile range). Significance was determined by one-way analysis of variance (ANOVA) with Bonferroni multiple-comparison analysis (**b–f**) and Pearson's correlations (**g**, **h**). * or # $P < 0.05$, ** or ## $P < 0.01$, *** or ### $P < 0.001$, NS. non-significance

every 2 weeks and were found similar between FGF21KO and WT mice fed with STC. FGF21KO mice had less body weight, especially fat mass, when comparing to the WT littermates from 6 weeks after HFD induction (Fig. 3a, b). However, the difference in lean mass and fluid between FGF21KO and WT were not significant after HFD induction (Supplementary Fig. 2). To further investigate the specific depot of fat in FGF21KO mice, various adipose depots were weighed, including inguinal SAT,

epididymal (epiVAT) and perirenal visceral adipose tissue (periVAT). Notably, SAT mass in FGF21KO mice was significantly lower than that in WT mice, but VAT mass was similar between WT and FGF21KO mice (Fig. 3c), suggesting that FGF21 is required for the accumulation of subcutaneous fat in diet-induced obesity.

To verify the roles of FGF21 in the accumulation of subcutaneous fat mass, we replenished FGF21KO mice with

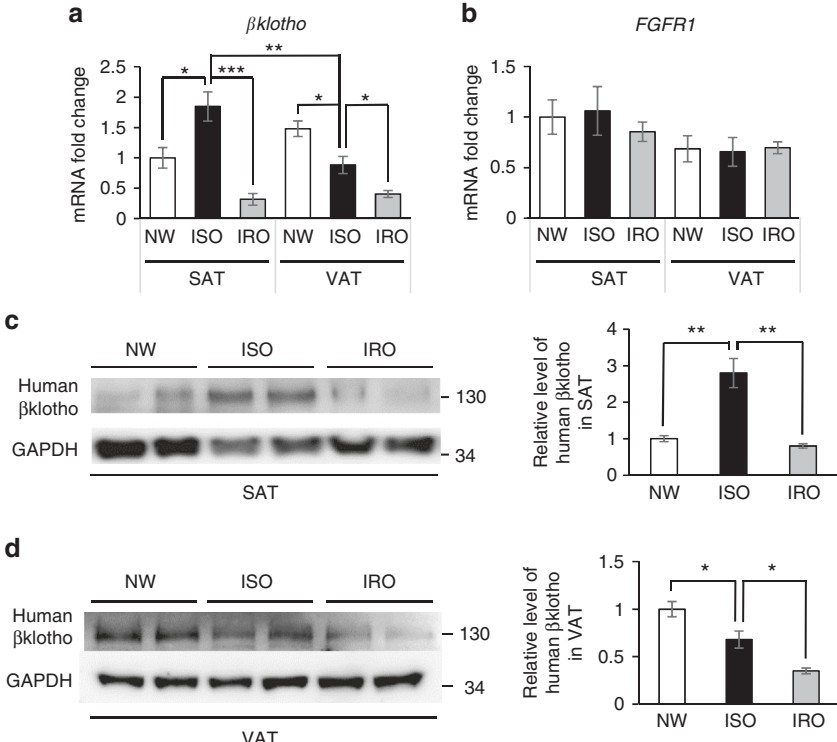

**Fig. 2** βklotho exhibits fat depot-difference in the development of obesity. **a** Change of βklotho expression in SAT and VAT of humans with NW, ISO and IRO. All data were normalized to βklotho expression in SAT of NW individuals. $n = 8–12$. **b** Change of FGFR1 expression in SAT and VAT of individuals with NW, ISO and IRO. All data were normalized to FGFR1 expression in SAT of NW individuals. $n = 8–12$. **c,d** Representitive figure of western blot analysis and densitometric quantification of human βklotho protein levels in (**c**) SAT and (**d**) VAT of individuals with NW, ISO and IRO ($n = 5$). Data are presented as mean ± s.e.m. Significance was determined by one-way ANOVA with Bonferroni multiple-comparison analysis. *$P < 0.05$, **$P < 0.01$, ***$P < 0.001$

recombinant mouse FGF21 (rmFGF21) to observe the change of fat distribution and other metabolic parameters. Of note, in this study, we used a physiological dose of rmFGF21 to mimic the HFD-induced FGF21 level in FGF21KO mice, which is different from previous studies using a pharmaceutical dose of rmFGF21 ($\geq 1$ mg kg$^{-1}$ day$^{-1}$) in obese mice[3,4]. Eight-week-old FGF21KO and WT mice were fed with HFD for 8 weeks, followed by replenishment with rmFGF21 (0.1 mg kg$^{-1}$ day$^{-1}$) or saline (as a vehicle control) with osmotic pumps for another 4 weeks (Fig. 3e). FGF21 levels in WT mice fed with HFD for 8 weeks were about 0.6 ng ml$^{-1}$ (Fig. 3d). In consideration of the bioactivity of recombinant protein, the rmFGF21 dose of 0.1 mg kg$^{-1}$ day$^{-1}$ could induce a stable FGF21 concentration to 1.5 ng ml$^{-1}$ in circulation, which was 2–3 times of the 8-week HFD-induced endogenous FGF21 level comparable to the endogenous FGF21 levels seen in the serum of HFD-fed mice (Fig. 3f). After continuous infusion for 4 weeks, the net weight and total fat mass did not change in FGF21KO mice (Supplementary Fig. 3a,b). However, the fat distribution was changed obviously. Compared with the FGF21KO + Vehicle group, SAT was significantly increased and VAT was slightly decreased in the FGF21KO + rmFGF21 group. These results were further validated by abdominal MRI in live mice and confirmed by weighing various fat depots dissected (Supplementary Fig. 3e and Fig. 3g, h). Histological analysis revealed that chronic treatment of rmFGF21 with a physiological dose increased the number of small-size adipocytes and decreased the number of large adipocytes in subcutaneous fat of FGF21KO mice (Fig. 3i). These results suggest that replenishment with rmFGF21 in FGF21KO mice to physiologically relevant HFD-induced levels restores

subcutaneous fat mass. FGF21-induced expansion of subcutaneous fat mainly results from hyperplasia of adipocytes.

Adipose tissue-specific βklotho knockout (Klb AdipoKO) mice exhibited modestly decreased subcutaneous fat after HFD induction (Fig. 4). After 8-week of HFD induction, we used a physiological dose of rmFGF21 by an osmotic pump (0.05 mg kg$^{-1}$ day$^{-1}$) to mimic HFD-induced circulating FGF21 level in Klb AdipoKO mice for another 4 weeks. The circulating FGF21 level was similar as the level during rmFGF21 replenishment in FGF21KO mice (Fig. 4e). However, unlike FGF21 KO mice, Klb AdipoKO mice were refractory to the effects of physiological concentrations of FGF21 on the expansion of subcutaneous fat (Fig. 4f). Taken together, this result further demonstrates that elevation of FGF21 in circulation and its action on adipose tissue lead to the accumulation of subcutaneous fat mass in diet-induced obesity.

**FGF21 increases insulin sensitivity via expansion of SAT**. It was revealed that FGF21KO mice and Klb AdipoKO mice exhibited exacerbated glucose intolerance and insulin resistance than WT mice after HFD induction (Figs. 5a–c, 4g–i). After 4 weeks of replenishment with rmFGF21 to a HFD-induced level, the impaired glucose tolerance and insulin sensitivity in FGF21KO mice were fully rescued (Fig. 5a–c). However, Klb AdipoKO mice are refractory to the effects of physiological concentrations of FGF21 on the alleviation of glucose intolerance and insulin resistance (Fig. 4g–i). These results further suggest that the metabolic benefits of FGF21 are mediated at least in part by the expansion of subcutaneous fat.

Systemic insulin sensitivity as quantified by GIR was significantly lower in FGF21KO mice than those in WT mice, and it was restored after rmFGF21 replenishment (Fig. 5d). During the clamp, insulin further decreased hepatic glucose production (HGP) in the FGF21KO + rmFGF21 group in comparison to their corresponding basal HGP levels (Fig. 5e). The uptake of 2-[14C]deoxyglucose (2-[14C]DG) into inguinal SAT per milligram was lower in FGF21KO mice in comparison to WT mice, and regained after rmFGF21 replenishment (Fig. 5f). Total glucose uptake in SAT of FGF21KO mice was reduced by

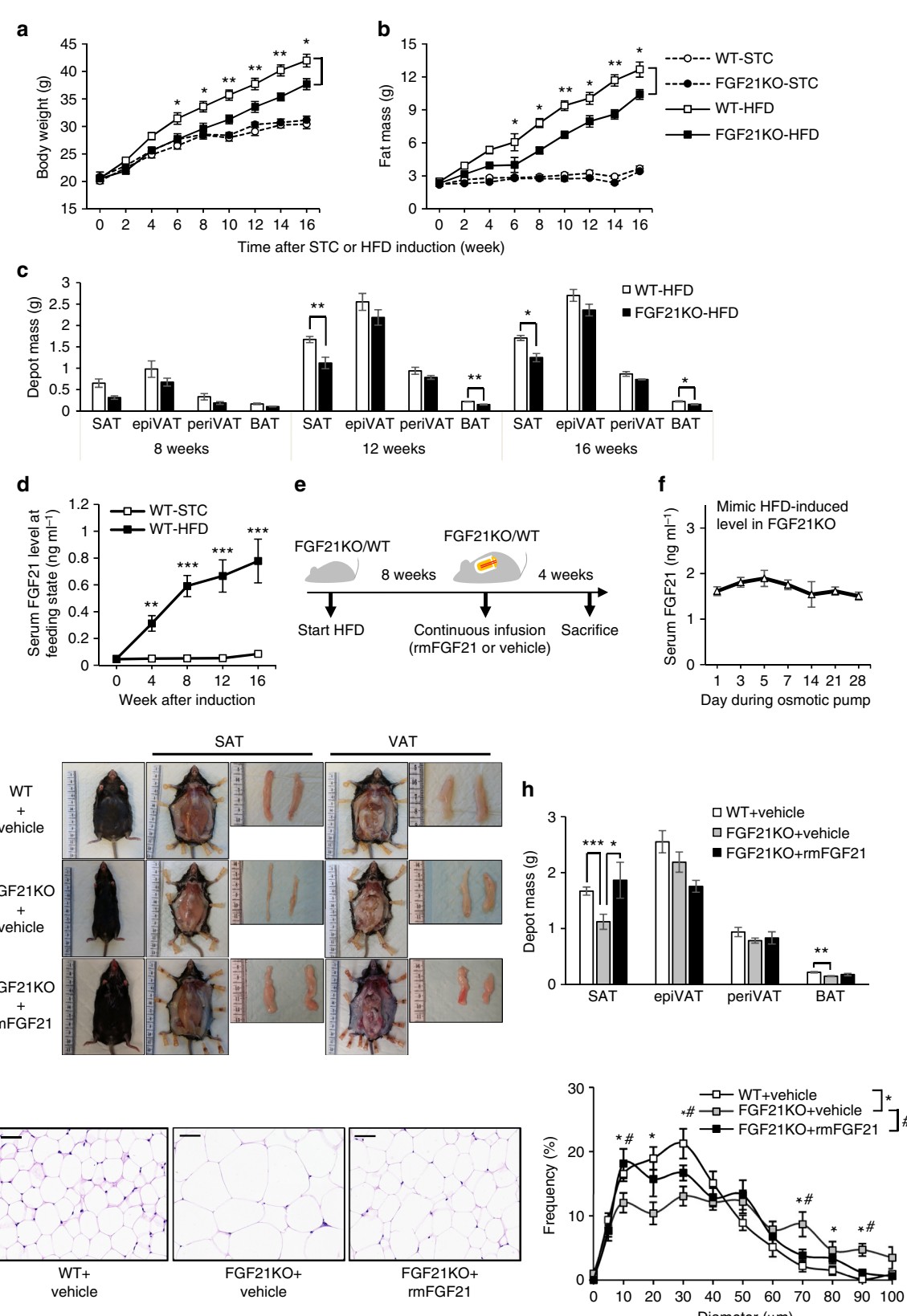

**Fig. 3** FGF21 is required for the accumulation of SAT mass in diet-induced obesity. Eight-week-old, male FGF21KO and WT littermates were fed with STC or HFD for 16 weeks (n = 10–12). **a, b** Body weight (**a**) and fat mass were measured once every 2 weeks (**b**). **c** Depot mass of SAT, epiVAT, periVAT and interscapular BAT depots in FGF21KO and WT mice fed with HFD. **d** Serum FGF21 levels at feeding state in WT mice during STC or HFD induction were measured by ELISA. **e** Schematic diagram of rmFGF21 replenishing strategy. After 8-week HFD induction, FGF21KO mice were randomly divided into FGF21KO + rmFGF21 group (0.1 mg kg$^{-1}$ day$^{-1}$ rmFGF21 by osmotic pump to mimic HFD-induced circulating FGF21 level) and FGF21KO + Vehicle group (receiving saline by osmotic pump) for another 4 weeks. The WT group also received continuous infusion of saline. n = 6. These results were reproduced in four independent experiments. **f** Serum FGF21 levels in FGF21KO + rmFGF21 group during the intervention. **g** Effect of physiologically-relevant dose of rmFGF21 administration on subcutaneous and visceral fat mass. Representative photographs of body appearance, subcutaneous fat and dissected SAT, visceral fat and dissected epiVAT from three groups. **h** Depot mass of SAT, epiVAT, periVAT and BAT depots in three groups. **i** H&E staining on paraffin sections from SAT in three groups. Scale bar = 50 μm. Cell size profiling of adipocytes from SAT was compared among three groups. The values show % from the total number of analyzed cells. Data are presented as mean ± s.e.m. Significance was determined by student's $t$ test (**c, d**), one-way ANOVA (**h, i**) and two-way ANOVA with Bonferroni multiple-comparison analysis (**a, b**). * or # $P < 0.05$, **$P < 0.01$, ***$P < 0.001$

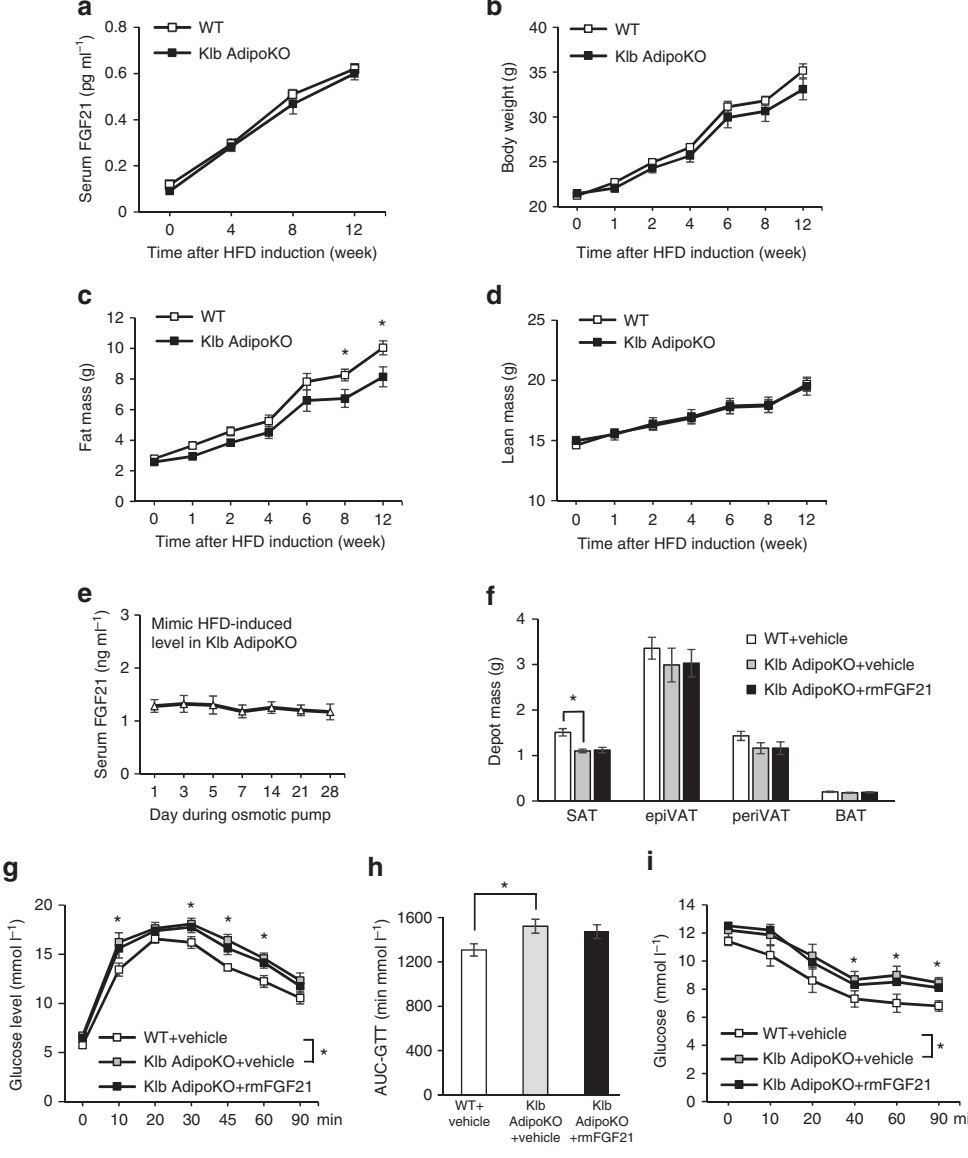

**Fig. 4** Klb AdipoKO mice do not benefit from physiological dose of rmFGF21. Eight-week-old, male WT and Klb AdipoKO mice were fed with HFD for 12 weeks. n = 8. **a** Serum FGF21 levels at fed state in WT and Klb AdipoKO mice during HFD induction were measured by ELISA. **b–d** Body weight (**b**), fat mass (**c**) and lean mass (**d**) were measured at various time periods. After 8-week HFD induction, Klb AdipoKO mice were randomly divided into Klb AdipoKO + rmFGF21 group (0.05 mg kg$^{-1}$ day$^{-1}$ rmFGF21 by osmotic pump to mimic HFD-induced circulating FGF21 level) and Klb AdipoKO + Vehicle group (receiving saline by osmotic pump) for another 4 weeks. The WT group also received continuous infusion of saline. n = 6. **e** Serum FGF21 levels in Klb AdipoKO + rmFGF21 group during the intervention. **f** Adipose tissue mass was measured for SAT, epiVAT, periVAT and BAT depots in three groups after 4 weeks of intervention. **g, h** Glucose tolerance test (GTT) (**g**) was performed and area under curve (AUC) analysis of GTT (**h**), Insulin tolerance test (ITT) (**i**) was performed in three groups after 4 weeks of intervention. Data are presented as mean ± s.e.m. Significance was determined by student's $t$ test (**a–d**), one-way ANOVA (**f,h**) and two-way ANOVA with Bonferroni multiple-comparison analysis (**g, i**). *$P < 0.05$

nearly 80% and substantially restored by replenishment of rmFGF21 (Fig. 5g). Consistently, FGF21KO mice were accompanied by a significant elevation in insulin-stimulated Akt phosphorylation (S473) in SAT after replenishment with rmFGF21 (Fig. 5k). The uptake of 2-[$^{14}$C]DG into other tissues,

including epiVAT, muscle and brown adipose tissue (BAT) were similar among three groups (Fig. 5h–j). Consistent with our previous reports, serum adiponectin in FGF21KO mice was significantly lower than those in WT mice after HFD induction (Fig. 5l). The reduction of adiponectin was found in both SAT

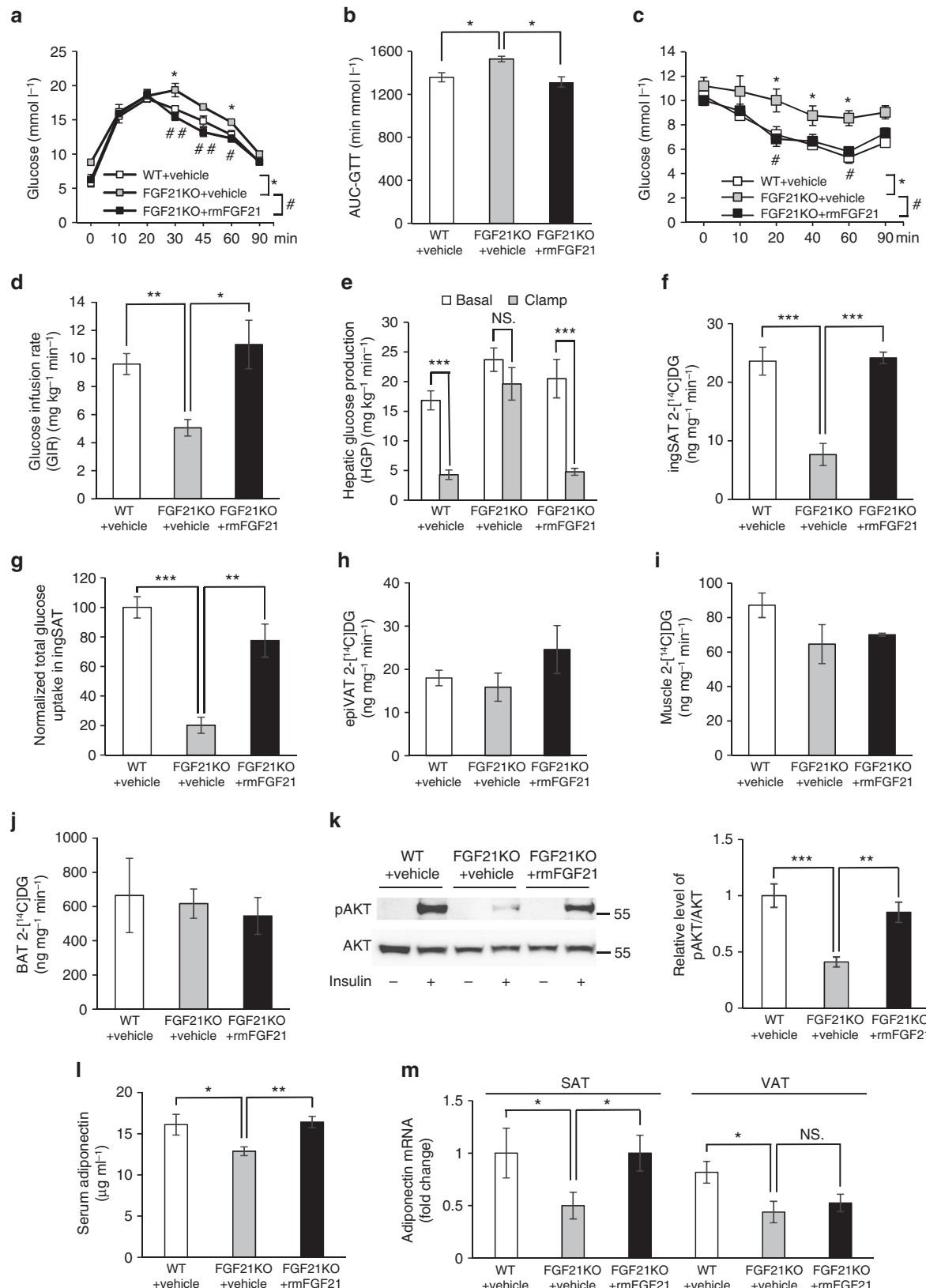

**Fig. 5** Physiological dose of rmFGF21 restores insulin sensitivity in FGF21KO mice. **a–c** GTT (**a**), AUC analysis of GTT (**b**) and ITT performed in WT+Vehicle (**c**), FGF21KO + Vehicle and FGF21KO+rmFGF21 groups after 4 weeks of intervention. $n = 6$. These results were reproduced in four independent experiments. *comparison of WT + Vehicle vs. FGF21KO + Vehicle, #comparison of FGF21KO + Vehicle vs. FGF21KO + rmFGF21. **d** Whole-body insulin sensitivity as quantified by GIR. $n = 6$. These results of hyperinsulinemic–euglycemic clamp were reproduced in two independent experiments. **e** HGP in the basal and clamp state. **f–j** Insulin-stimulated 2-[$^{14}$C]DG uptake in inguinal SAT (**f**, **g**), epiVAT (**h**), muscle (**i**) and BAT among three groups after 4 weeks of intervention (**j**). **k** Immunoblot of in vivo insulin-stimulated SAT after replenishment with rmFGF21 to HFD-induced level in FGF21KO mice. Representative immunoblots showing phospho (S473) and total Akt in the SAT of three groups. The tissues were collected at 10 min after single i.v. injection of insulin (1 U kg$^{-1}$) in mice. $n = 6$. The bar chart shows the densitometric analysis of phosphorylation levels. **l, m** Change of serum adiponectin (**l**) and adiponectin expression in different fat depots (SAT and VAT) among WT+Vehicle, FGF21KO+Vehicle and FGF21KO +rmFGF21 groups after 4 weeks of intervention were measured by ELISA and RT-qPCR (**m**). $n = 6$. These results were reproduced in four independent experiments. Data are presented as mean ± s.e.m. Significance was determined by one-way ANOVA (**b,d–m**) and two-way ANOVA with Bonferroni multiple-comparison analysis (**a, c**). * or # $P < 0.05$, ** or ## $P < 0.01$, ***$P < 0.001$, NS non-significance

and VAT in FGF21KO mice (Fig. 5m). Serum adiponectin levels in FGF21KO mice were restored after replenishment with rmFGF21 for 4 weeks (Fig. 5l). On tissue level, adiponectin expression in SAT was completely restored after rmFGF21 replenishment but adiponectin expression in VAT was unchanged (Fig. 5m). These results indicate that subcutaneous fat mass is an important contributor to the effect of FGF21 on systemic insulin sensitivity.

As shown in Fig. 3h, the crude weight of BAT in FGF21KO mice did not show any significant change after the replenishment with rmFGF21 to a physiologically-relevant HFD-induced level. There is no obvious difference in either relative UCP-1 protein level or total UCP-1 level per BAT depot (as determined by multiplying the UCP-1 protein level per mg homogenate protein with the total amount of proteins in each depot[29]) among the three groups (Supplementary Fig. 4a–c). Furthermore, replenishment with physiological dose of rmFGF21 did not affect the protein levels of UCP-1 in SAT (Supplementary Fig. 4d). These results suggest that the effect of physiological level of FGF21 in mediating glucose metabolism is independent of the activation of BAT and browning of SAT.

**SAT transplantation improves insulin sensitivity of KO mice**. To further explore the role of FGF21 in glucose homeostasis via subcutaneous fat, we used a fat transplantation strategy. After feeding with HFD for 8 weeks, a total of 0.85 g subcutaneous fat from FGF21KO or WT donor mice was transplanted into the inguinal area of FGF21KO host mice (KO→KO and WT→KO respectively) (Fig. 6a). The FGF21KO Sham and WT Sham groups had surgery in the same area, but no fat was transplanted. The viability of the fat graft was confirmed by vascularization and normal morphology of adipocytes (Fig. 6b, c). Eighteen days after fat transplantations, glucose tolerance and insulin sensitivity were partially restored in WT→KO group (Fig. 6d–g). Hepatic insulin sensitivity and glucose uptake in endogenous SAT of FGF21KO mice were improved by transplantation of subcutaneous fat from WT mice (Fig. 6h,i). 2-[$^{14}$C]DG uptake into other tissue including epiVAT, muscle and BAT were unchanged after fat transplantation (Fig. 6j–l). The improvement of glucose metabolism did not occur in KO→KO group. Consistently, FGF21KO mice were accompanied by a significant elevation in insulin-stimulated Akt phosphorylation (S473) in endogenous SAT after fat transplantation from WT mice (Fig. 6m). Moreover, serum adiponectin levels in FGF21KO mice were partially restored in WT→KO group (Fig. 6n). On tissue level, the expression of adiponectin was also partially restored in SAT but not in VAT (Fig. 6o). Serum FGF21 levels remained undetectable in FGF21KO mice transplanted with subcutaneous fat from WT donor mice. These results suggest that subcutaneous fat transplantation may exert its metabolic benefits in FGF21KO mice in part by promoting the release of adiponectin.

**FGF21 specifically modulates biogenesis and function of SAT**. To investigate how FGF21 functions to specifically enlarge subcutaneous fat, we studied the fat-depot difference of the genes involved in adipogenesis and insulin signaling in HFD-induced FGF21KO and Klb AdipoKO mice. We found that FGF21KO mice have reduced expressions of *Cebpa*, *Srebf1a*, *Srebf1c*, *InsR*, *IRS-1*, phosphatidylinositol 3-kinase (*PI3K*) and glucose transporter 4 (*GLUT4*) in SAT after HFD induction, but such changes were not found in VAT (Fig. 7a). The reduced expressions of the genes involved in adipogenesis and insulin signaling in SAT were also observed in HFD-induced Klb AdipoKO mice (Fig. 7b). After replenishment with rmFGF21, expressions of these genes in SAT were partially restored in FGF21KO mice (Fig. 7c). The elevation in insulin-stimulated Akt phosphorylation (S473) in SAT after replenishment with rmFGF21 was confirmed by immunoblot analysis (Fig. 5k). These results suggest that *FGF21* deficiency causes reduced expressions of the genes involved in adipogenesis and insulin signaling especially in SAT in diet-induced obesity.

To determine whether the adipose tissue phenotype was due to their distinct properties, we examined the differentiation of preadipocytes derived from SAT of WT, FGF21KO and Klb AdipoKO mice (Supplementary Fig. 5). The induction of *Cebpa* and *Pparg2* mRNAs, which play important roles in the early stages of adipocyte differentiation, was delayed in FGF21KO and Klb AdipoKO mice adipocytes. *Pparg2* and *Cebpa* expressions were efficiently restored by rmFGF21 in FGF21KO but not in Klb AdipoKO adipocytes (Supplementary Fig. 5a). The expressions of *InsR*, *IRS-1* and *PI3K* were efficiently restored by rmFGF21 in FGF21KO but not in Klb AdipoKO differentiated adipocyte (Supplementary Fig. 5b). Treatment with rmFGF21 had a minimal effect or almost no effect on the adipogenesis and lipid accumulation of primary adipocytes converted from VAT (Supplementary Fig. 5c). These results suggest that FGF21 modulates biogenesis and insulin sensitivity of SAT, during such progress the up-regulation of βklotho is indispensable.

**FGF21 promotes M2 macrophage polarization in SAT**. In the stage of obesity with retained insulin sensitivity, M2-polarized resident adipose tissue macrophages (ATMs) are able to partially protect adipocytes from these inflammatory factors and may block M1 polarization[30]. Thus, we investigated whether FGF21 involved in macrophage polarization that may protect adipocytes from inflammation which contributes to insulin resistance. We found that FGF21KO mice had decreased macrophages in SAT after HFD induction as revealed by the number of F4/80$^+$cells and rmFGF21 replenishment led to a dramatic enrichment of macrophages in SAT (Fig. 8a; Supplementary Fig. 6). Moreover, flow cytometry analysis showed that rmFGF21 replenishment in FGF21KO mice was accompanied by increased composition of M2 macrophages (cd11c$^{low}$cd206$^{high}$) while the composition of M1 macrophages (cd11c$^{high}$cd206$^{low}$) remained constant (Fig. 8a,

b). Consistently, quantitative real-time polymerase chain reaction (PCR) analysis demonstrated that *F4/80* and markers of M2 macrophages including *Arg1*, *Mgl1*, *Mgl2*, *MRC2* and *IL10* were decreased in FGF21KO mice and were partially restored upon rmFGF21 replenishment. However, markers of M1 macrophages

including *cd11c*, *MCP1*, *TNFα*, *IL1β* and *IL6* were unchanged (Fig. 8c). In contrast to the increase of M2 macrophages in SAT, such changes were not observed in VAT (Supplementary Fig. 7a). Eighteen days after fat transplantations, number of F4/80+cells and M2 macrophages were at least partially restored in SAT of

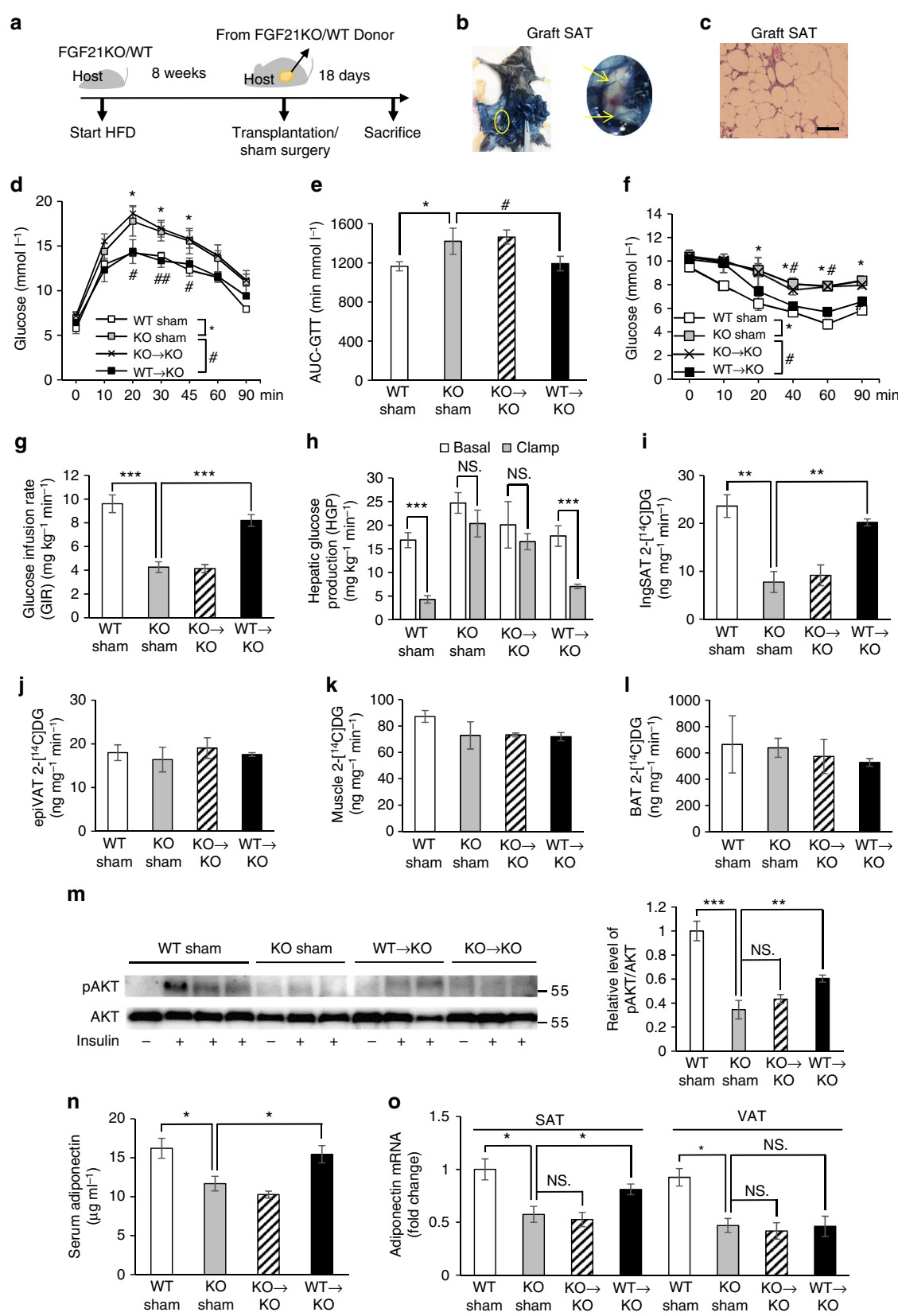

**Fig. 6** Transplantation of SAT from WT to FGF21KO mice improves insulin sensitivity. **a** Schematic diagram of transplantation strategy. After 8-week HFD induction, a total of 0.85g-subcutaneous fat from WT or FGF21KO donor mice was transplanted into the inguinal area of FGF21KO host mice (KO→KO and WT→KO groups). WT Sham and FGF21KO Sham groups had surgery in the same area, but no fat was transplanted. Eighteen days after fat transplantations, various measurements were carried out. $n = 6$. These results were reproduced in three independent experiments. **b** Evans blue i.v. injection into host mice to prove vascularization of the fat grafts at 2 weeks after transplantation. **c** Hematoxylin and eosin-stained fat grafts showed normal morphology of adipocytes. Scale bar = 50 μM. **d–f** GTT (**d**), AUC analysis of GTT (**e**) and ITT performed in all four groups including WT Sham, KO Sham, KO→KO and WT→KO (**f**). *comparison of WT Sham, vs. KO Sham, #comparison of KO Sham vs. WT→KO. **g** Whole-body insulin sensitivity as quantified by GIR. **h–l** HGP in the basal and clamp state (**h**), insulin-stimulated 2-[14C]DG uptake in inguinal SAT (**i**), epiVAT (**j**), muscle (**k**) and BAT among all four groups (**l**). (**m**) Immunoblot of in vivo insulin-stimulated SAT in transplantation cohort. $n = 6$. Representative immunoblots showing phospho (S473) and total Akt in the endogenous SAT of all four groups. The tissues were collected at 10 min after single i.v. injection of insulin (1 U kg$^{-1}$). The bar chart shows densitometric analysis of phosphorylation levels. **n,o** Change of serum adiponectin (**n**) and adiponectin expression in different fat depots (SAT and VAT) in the WT Sham, KO Sham, KO→KO and WT→KO groups at 18th day after fat transplantation (**o**). $n = 6$. These results were reproduced in three independent experiments. Data are presented as mean ± s.e.m. Significance was determined by Student's $t$-test analysis (**h**), one-way ANOVA (**e**, **g**, **i–o**) and two-way ANOVA with Bonferroni multiple-comparison analysis (**d**, **f**). * or # $P < 0.05$, ** or ## $P < 0.01$, ***$P < 0.001$, NS non-significance

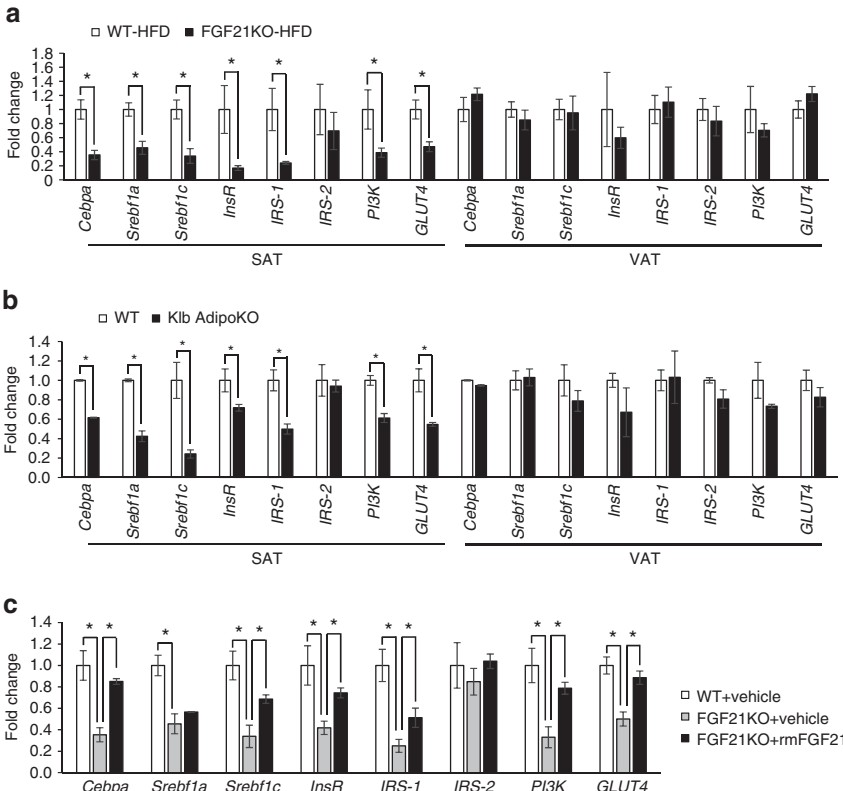

**Fig. 7** FGF21KO and Klb AdipoKO mice have altered gene expressions in SAT. **a** The expression of the genes involved in adipogenesis and insulin signaling in SAT and VAT of WT and FGF21KO mice fed on HFD for 8 weeks. All the mice were killed during the fed state. $n = 8$–10. **b** Related gene expression in the SAT and VAT of WT and Klb AdipoKO mice fed with HFD for 8 weeks. All the mice were killed during the fed state. $n = 8$. **c** Related gene expression in SAT among WT + Vehicle, FGF21KO + Vehicle and FGF21KO + rmFGF21 groups after 4 weeks' intervention. $n = 6$. These results were reproduced in four independent experiments. Data are presented as mean ± s.e.m. Significance was determined by student's $t$ test (**a**, **b**) and one-way ANOVA with Bonferroni multiple-comparison analysis (**c**). *$P < 0.05$

WT→KO group but not in KO→KO group (Fig. 8d–f). Likewise, such changes were not observed in VAT (Supplementary Fig. 7b). These data suggested that FGF21 promoted M2 macrophage polarization in subcutaneous fat. The in-vitro study found that anti-inflammatory cytokine IL10, which was elevated during M2 macrophage polarization, prevented the effects of TNFα on blocking insulin-stimulated glucose uptake in differentiated stromal vascular fraction (SVF) adipocytes from FGF21KO mice (Fig. 8g). These data suggest that M2 macrophage polarization and its anti-inflammatory changes promoted by FGF21 in sub-cutaneous fat may protect adipocytes from proinflammatory state

and act as compensatory responses to maintain insulin sensitivity in diet-induced obesity.

## Discussion

A growing body of evidence suggests that subcutaneous fat is protective whereas visceral fat is detrimental to metabolic health[19–25,31]. A recent large epidemiological study further highlights that subcutaneous fat is protective of cardiometabolic diseases and mortality in the European population[32]. However, how these two kinds of fat depots are differentially regulated

remains poorly understood. In this study, we provide both animal and human evidence demonstrating for the first time that the metabolic hormone FGF21 enhances insulin sensitivity by regulating the selective expansion of subcutaneous fat (Fig. 9). In animals, we demonstrate that the decreased insulin sensitivity in FGF21KO mice is associated with reduced subcutaneous fat,

whereas the replenishment of rmFGF21 to a level similar to those occurring in diet-induced obesity can reverse insulin resistance as well as increase the amount of subcutaneous fat. Furthermore, transplantation of subcutaneous fat into FGF21KO mice can reverse their diet-induced insulin-resistant phenotype. In humans, we found an independent association between FGF21

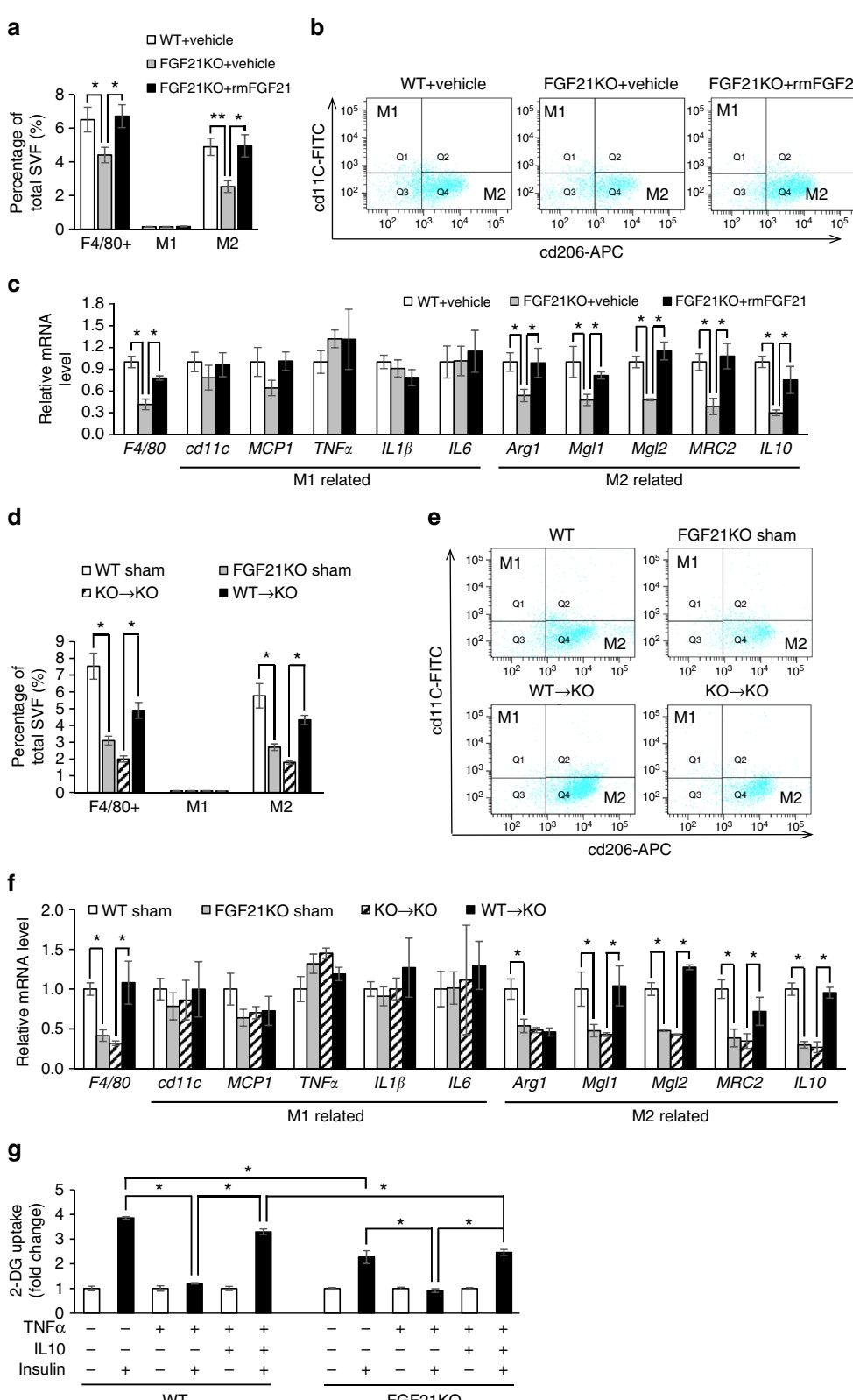

**Fig. 8** FGF21 promotes M2 macrophage polarization specifically in SAT. **a–c** After 8-week HFD induction and 4 weeks of treatment by vehicle or physiologically relevant dose of rmFGF21 (0.1 mg kg$^{-1}$ day$^{-1}$), another cohort of WT + Vehicle, FGF21KO + Vehicle and FGF21KO + rmFGF21 mice were killed for flow cytometry and other following experiments. **a** Flow cytometry analysis for total, M1 and M2 macrophages in SVF of SAT. **b** Representative plot chats of flow cytometry analyzing M1 and M2 macrophages after gating F4/80$^+$ cells in SAT. M2 macrophages were defined as F4/80$^+$cd11c$^{low}$cd206$^{high}$ and M1 macrophages were defined as F4/80$^+$cd11c$^{high}$cd206$^{low}$. The gating strategy is described in Supplementary Fig. 6a. **c** RT-qPCR analysis for mRNA expression levels of F4/80, M1 and M2 macrophages related genes. $n = 6$. These results were reproduced in two independent experiments. (**d–f**) At 18th day after fat transplantation, another cohort of WT Sham, KO Sham, KO→KO and WT→KO mice were killed for flow cytometry and other following experiments. **d** Flow cytometry analysis for total, M1 and M2 macrophages in SVF of SAT. **e** Representative plot chats of flow cytometry analyzing M1 and M2 macrophages in SAT. **f** RT-qPCR analysis for mRNA expression levels of F4/80, M1 and M2 macrophages related genes. $n = 6$. These results were reproduced in two independent experiments. **g** Insulin-stimulated glucose uptake in adipocytes chronically treated with interleukin (IL)-10 and tumor necrosis factor (TNF)-α. Differentiated SVF adipocytes from WT and FGF21KO mice were treated with low-dose TNFα (3 ng ml$^{-1}$) for 72 h in the presence or absence of IL10 (20 ng ml$^{-1}$). Without (white bars) or with insulin (black bars) (100 nm) stimulation, 2-DG uptake was assessed. $n = 6$ wells. These results were reproduced in three independent experiments. Data are presented as mean ± s.e.m. Significance was determined by one-way ANOVA with Bonferroni multiple-comparison analysis. *$P < 0.05$, **$P < 0.01$

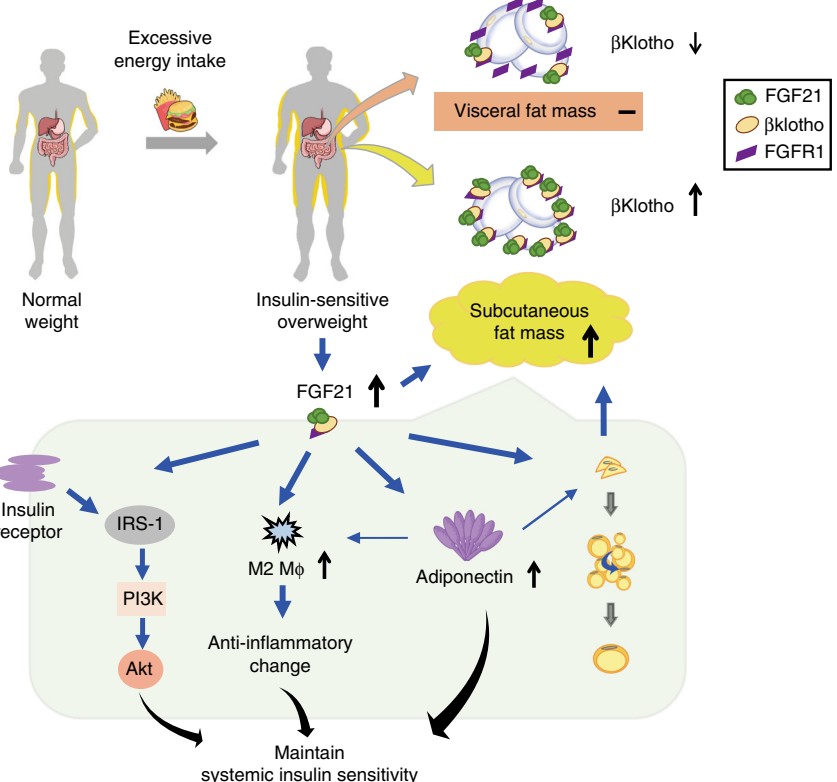

**Fig. 9** Proposed action of FGF21 on systemic insulin sensitivity by expansion of SAT. Due to excessive energy intake, individuals with normal weight can develop insulin-sensitive overweight featured by the expansion of subcutaneous fat mass. Circulating FGF21 acts in an endocrine manner on subcutaneous fat to promote the healthy expansion. Such an adipose-specific action of FGF21 is attributed to high level of expression of the FGF21 receptor complex. FGF21 upregulates adiponectin in subcutaneous fat, consistent with an increase of M2 macrophage polarization in the subcutaneous adipocytes and the following anti-inflammatory change. These factors maintain systemic insulin sensitivity. Elevated endogenous FGF21 in response to dietary-induced obesity serves as a defense mechanism against systemic insulin resistance

levels and the amount of subcutaneous fat in insulin-sensitive obese individuals.

Our present study first demonstrated that serum FGF21 levels in individuals with ISO were positively associated with their SFA but not VFA. The change of circulating FGF21 in insulin-sensitive overweight was more remarkable than adiponectin, which is regarded as a potent factor that increases the metabolic flexibility of adipose tissue[33]. Consistently, we provide evidence from both rodents and humans showing that βklotho levels exhibit fat depot-difference in diet-induced obesity. A previous study found that βklotho levels were reduced in visceral fat of HFD-induced obese mice through the suppression by TNFα[34]. Yet in peripheral obesity when insulin sensitivity is preserved, we

observed the increase of βklotho level in subcutaneous fat but the decrease in visceral fat, suggesting that FGF21 preserves insulin sensitivity via its action on subcutaneous fat. Consistent with our findings, a recent study on monkeys chronically fed with HFD demonstrated that HFD-resistant monkeys (remaining insulin-sensitive after HFD) exhibited increased β-klotho levels in subcutaneous fat compared to HFD-sensitive monkeys[35]. By contrast, another study showed that βklotho level was significantly decreased in subcutaneous fat in obese individuals with normal glucose level[36]. However, the BMI of obese individuals with normal glucose level was $39.2 \pm 1.2$ kg m$^{-2}$ and their average HOMA-IR was $3.3 \pm 0.3$. Therefore, these individuals are insulin-resistant and belong to the IRO group in our study[36].

We demonstrate that SAT but not VAT is the major target of FGF21 in diet-induced obesity. Consistent with our study, evidence exists indicating that adipose tissue is the primary target in mediating the metabolic action of FGF21[37,38]. Both FGF21 receptor *FGFR1* and coreceptor *βklotho* are highly expressed in adipose tissue in both humans and animals[37]. Furthermore, the insulin-sensitizing actions of FGF21 are abrogated in lipodystrophic mice and mice with adipose-specific ablation of *βklotho*[37,38]. Conversely, transplantation of adipose tissue into the lipodystrophic mice restores the FGF21 responsiveness, indicating that adipose tissue is an indispensable mediator of FGF21 functions[38].

Previous studies revealed unremarkable metabolic phenotypes in young and STC diet-fed FGF21KO mice[39–41]. One of the FGF21KO lines was shown to be lipodystrophic[42]. Our present study found that FGF21KO mice had less body weight especially less subcutaneous fat mass compared to the WT littermates after HFD induction. Pharmacological administration of recombinant FGF21 results in multiple metabolic benefits, including reduced body weight, easing glycaemia by promoting peripheral utilization and decreasing the hepatic production of glucose as well as decreasing triglycerides and low-density lipoprotein cholesterol[43]. In the present study, an important emphasis was placed on replenishing FGF21KO mice with rmFGF21 to a level similar to those occurring in diet-induced obesity and obvious change of fat distribution was observed. Adipogenesis of subcutaneous fat depot was specifically promoted and subcutaneous fat mass was enlarged, together with the improvement of glucose tolerance. We found that FGF21 adipose-specific knockout mice (FGF21 AdipoKO) did not have reduced circulating FGF21 level during HFD feeding, and their fat distribution was similar to those of WT littermates (Supplementary Fig. 8), which is consistent with the previous finding[44]. We propose that circulating FGF21 acts in an endocrine manner on subcutaneous fat to promote the healthy expansion, which in turn produces other factors to promote systemic insulin sensitivity.

Some genetically engineered mouse models - such as Mitoneet, *aP2*-Glut4 or adiponectin transgenic animals[33,45,46]—exhibit a "healthy-obese" phenotype. This obese phenotype includes preferential expansion of SAT, and unimpaired insulin sensitivity and glucose tolerance. The inability to appropriately expand the SAT is one of the key factors that link excess caloric intake to insulin resistance[33]. PPARγ agonist, thiazolidinedione, improves insulin sensitivity despite increasing the subcutaneous fat mass[47]. Both data obtained in the genetically engineered and pharmacological models directly highlights that subcutaneous fat mass expansion has potent antidiabetic effects. Using an in vivo fat transplantation strategy, SAT has been shown to have direct and beneficial effects on control of body weight and metabolism[31]. Another study using fat transplantation showed that mice receiving SAT from exercise-trained mice had improved glucose tolerance and enhanced insulin sensitivity compared to mice transplanted with SAT from sedentary mice[48]. These effects appear to be due to a distinct property rather than anatomic location of the subcutaneous fat, most likely secretion of one or more factors that can mediate improvements in the metabolic profile. This notion is supported by our observation that subcutaneous fat transplantation in FGF21KO mice results in changes in the release of adiponectin, which may contribute to the systemic insulin sensitivity.

Apart from being able to generate beige adipose tissue, subcutaneous fat has been demonstrated to combat insulin resistance via multiple mechanisms[49–53]. Several studies have demonstrated that subcutaneous fat can secrete more adiponectin and leptin than visceral fat[49–51]. It has been reported that adiponectin and leptin genes were expressed at 10,000 to a million-fold higher in peripheral fat depots as compared to visceral fat depot in mice[49]. In humans, total adiponectin, as well as leptin, were higher in individuals with more subcutaneous fat than those with more visceral fat accumulation[50,51]. Moreover, compared to visceral fat, subcutaneous fat secretes less inflammatory cytokines[52,53]. The release of IL-6 was 2–3 times lower in SAT than that in omental adipose tissue[52]. It was also found that visceral but not subcutaneous fat was associated with increasing levels of C-reactive protein and TNFα[53]. Although the reason why subcutaneous fat is less prone to develop inflammation is unclear yet, one possible explanation is due to the protein inhibitor of activated STAT 1 (PIAS1) which is highly expressed in subcutaneous fat depot[54,55]. It was found that PIAS1 suppressed the inflammation in subcutaneous fat via inactivation of c-Jun N-terminal kinase[54,55]. These mechanisms might provide explanations to the benefits of the expansion of subcutaneous fat caused by physiologically-relevant HFD-induced FGF21 although browning of SAT is not affected.

Although the precise mechanisms whereby FGF21-mediated expansion of subcutaneous fat improves systemic insulin sensitivity remain to be defined, our data suggest that the release of adiponectin from subcutaneous fat is a contributor. It has been shown that FGF21 stimulates the production of adiponectin at both transcriptional and post-translational levels possibly via activation of PPARγ[40]. Our study revealed that replenishment with rmFGF21 and receiving transplantation of subcutaneous fat partially restored circulating adiponectin level in FGF21KO mice. Moreover, both elevations were mainly due to the increase of adiponectin production, especially in subcutaneous fat. This finding indicates that adiponectin is partially responsible for the missing link between subcutaneous fat and systemic insulin sensitivity. Adiponectin was reported to enhance hepatic insulin sensitivity by increasing insulin receptor substrate-2 expression and promoting the activation of AMPK in liver[56,57]. It was also reported that globular adiponectin can enhance glucose uptake in adipocytes via the activation of AMPK[58]. Additionally, the effects of FGF21 on promoting M2 macrophage polarization and production of anti-inflammatory cytokines in subcutaneous fat may also lead to the improvement of systemic insulin sensitivity. Diet-induced obesity leads to a shift in the activation state of ATMs from an M2-polarized state that may protect adipocytes from inflammation to an M1 proinflammatory state that contributes to insulin resistance[30]. In the present study, we showed that FGF21 promoted M2 macrophage polarization and M2-related anti-inflammatory cytokine such as IL10 in subcutaneous fat, which may act as a compensatory response to maintain glucose homeostasis in diet-induced obesity.

Taken together, our work provides evidence that elevated endogenous FGF21 in obesity serves as a defense mechanism against systemic insulin resistance. Furthermore, we have identified that FGF21 is a hormonal factor that regulates the selective expansion of the subcutaneous white fat depot. Our work provides mechanistic insights how subcutaneous fat acts as an endocrine tissue to alleviate systemic insulin resistance. Our study raises the possibility that targeting subcutaneous fat expansion through manipulation of FGF21 may represent a therapeutic strategy to combat insulin resistance and type 2 diabetes.

## Methods

**Clinical study.** The overweight and obese participants of this study are part of a cohort of individuals taking part in a longitudinal study aimed at investigating the pathophysiology of overweight and obesity. To be eligible for the present study, individuals had to have a BMI ≥ 25 kg m$^{-2}$, without T2DM and be off medications that may affect glucose metabolism, be otherwise healthy, and had a sedentary lifestyle. All participants who were enrolled in our cohort had a detailed medical history, a complete physical examination including a standard oral glucose tolerance test to determine carbohydrate tolerance and a dual-energy x-ray

absorptiometry to assess body composition. In these individuals with overweight or obesity, insulin-sensitive is defined as HOMA-IR <2.5 and insulin-resistant is defined as HOMA-IR ≥2.5[26]. Thirty ISO individuals were pair matched to 30 IRO individuals. Pair matching was based on similarities of age, gender, BMI, lean body mass, total fat mass and total fat percentage. Age and gender-matched participants with NW were recruited for this study through putting up posters in Shanghai Diabetes Institute. All individuals underwent comprehensive physical examinations and routine biochemical analyses of blood. Individuals with following conditions were excluded from this study: biliary obstructive diseases, acute or chronic virus hepatitis, cirrhosis, known hyperthyroidism or hypothyroidism, presence of cancer, current treatment with systemic corticosteroids and pregnancy.

Measurement of SFA and VFA was determined by MRI using a whole body imaging system (SMT-100, Shimadzu) with spin echo sequences: 500/20 (TR/TE) and matrix size = 256 × 256[59]. Scan time were approximately 180 s. MRI scans were obtained at the abdominal level between L4 and L5 vertebrae in the prone position. Analysis of the images was performed on a workstation provided by the manufacturer. MRI was performed by experienced radiologists who were blinded to clinical presentation and laboratory findings. Acquired images underwent measurement of SAT and VAT using a semiautomated segmentation method. According to the signal intensity of adipose tissue, SFA and VFA outline was manually traced with a graphic user interface. The area inside the outline was automatically labeled and calculated.

Insulin sensitivity was assessed by hyperinsulinemic–euglycemic clamp. Baseline samples were taken using iv catheters inserted into antecubital veins of both arms. A 10-min of the pretreatment period infusion of insulin (40 U ml$^{-1}$, Novo Nordisk) was initiated to raise the circulating insulin level before a constant infusion rate (40 mU m$^{-2}$ min$^{-1}$) was given to maintain the steady-state insulin at approximately 100 μU ml$^{-1}$ during the next 120 min. Glucose (20%) infusion was initiated simultaneously and continued at a variable dose to maintain plasma glucose concentration to the baseline level (Supplementary Fig. 9a). Blood samples were taken every 5 min and every 10 min for glucose and insulin measurement, respectively. The average GIR during 90–120 min was used as a measure of insulin sensitivity.

The demographic data and detailed medical histories were obtained using a standardized questionnaire. Plasma glucose was measured using the glucose oxidase method (Roche) and biochemical indexes were determined using an auto-analyzer (Hitachi 7600). Plasma levels of FGF21 and adiponectin were quantified using the enzyme-linked immunosorbent assay (ELISA) kits from Antibody and Immunoassay Services, the University of Hong Kong (AIS, HKU).

The adipose tissues were previously collected from another cohort of the study. These individuals with NW, ISO and IRO underwent laparotomy operation because of benign diseases (hepatic cyst, myoma of uterus, et al.) at the Department of Surgery. The samples were immediately shock-frozen and stored at −80 °C. The human study protocol was approved by the Human Research Ethics Committee and Review Board of the Shanghai Jiao Tong University School of Medicine, following the principles of the declaration of Helsinki. Written informed consent was obtained from all individuals.

**Animals**. Male FGF21KO mice generated by Department of Genetic Biochemistry, Kyoto University[40] and WT littermates with the same genetic background were used for this study. Klb AdipoKO mice were generated by crossing the *βklotho* gene floxed mice (Shanghai Nanfang Centre for Model Organisms) with adiponectin-Cre mice (The Jackson Laboratory, stock No. 010803). FGF21 AdipoKO mice were generated by crossing the *FGF21* gene floxed mice (Shanghai Nanfang Centre for Model Organisms) with aP2-Cre mice (The Jackson Laboratory, stock No. 005069)[60]. Both Klb AdipoKO and FGF21 AdipoKO mice were back-crossed with C57BL/6 J background for at least eight generations to ensure the genetic homogeneity. Klb AdipoKO and FGF21 AdipoKO mice were confirmed by genotyping and western blot analysis (Supplementary Fig. 10). No randomization of mice was used. These mice were fed with either STC or HFD (Research Diet, containing 45% fat, 20% protein, and 35% carbohydrate [kcal%]). All animals were kept under 12 h light-dark cycles at 22–24 °C, 60–70% humidity with free access to water. The sample sizes of animal experiments were estimated according to previous studies and the known variability of the assays. All animal experimental protocols were approved by the Animal Ethics Committee of the University of Hong Kong.

**MRI in mice**. Eight-week male FGF21KO mice and WT littermates fed with HFD for 8 weeks were replenished with rmFGF21 or vehicle for 4 weeks. MRI measurements of mice were performed utilizing a 7 T MRI scanner (70/16 PharmaScan, Bruker) by a radiologist who was blinded to group allocation. The mice were anesthetized with 3% isoflurane. During MRI, the animals were placed on a plastic cradle with the head fixed with a tooth bar and plastic screws in the ear canals. Continuous physiological monitoring was performed using an MRI-compatible system (SA Instruments). Spin echo sequences: TR/TE = 150/5.8 ms, FOV = 31.50 × 20.25 mm$^2$, matrix size = 210 × 135. The whole abdomen of each mouse were covered with axial slices (thickness 1 mm, no spacing). Acquired images underwent measurement of SAT and VAT using a semiautomated segmentation method. According to the signal intensity of adipose tissue, SFA and VFA outline was manually traced with a graphic user interface. The area inside the outline was automatically labeled and calculated.

**Replenishment with rmFGF21 at physiological doses**. Male FGF21KO mice and WT littermates (8-week-old) were fed with HFD for 8 weeks. To mimic the physiologically-relevant HFD-induced FGF21 level, FGF21KO mice were implanted with osmotic pumps (Alzet) on the back, slightly posterior to the scapulae to continuously deliver rmFGF21 (AIS, HKU) for 4 weeks (0.1 mg kg$^{-1}$ day$^{-1}$). Mice of FGF21KO + Vehicle and WT + Vehicle groups were implanted with an osmotic pump which delivered saline. The mice were first anesthetized and the skin over the implantation site were shaved and washed. A mid-scapular incision on the skin was made and a hemostat was inserted into the incision. The jaws were opened and closed to spread the subcutaneous tissue to create a pocket for the pump. Then a filled pump was put into the pocket, delivery portal first but not rest immediately beneath the incision. Close the wound with wound clips or sutures.

Circulating FGF21 levels were measured in FGF21 + rmFGF21 group during continuous infusion. Quantification of body composition and fat distribution, glucose tolerance and insulin tolerance tests were performed after 4 weeks of treatment. Mice were killed for collection of blood sample and various tissues for further biochemical evaluations. Another two cohorts were used for hyperinsulinemic–euglycemic clamp studies and flow cytometry analysis.

**Fat transplantation**. Eight-week male WT or FGF21KO donor mice fed with HFD for 8 weeks were anesthetized and subcutaneous fat was immediately removed, cut into approximately 0.2 g slices, and kept in saline on ice until transplantation. Eight-week male FGF21KO recipient mice fed with HFD for 8 weeks were anesthetized and a total of 0.85 g subcutaneous fat from WT or FGF21KO donor mice was transplanted into the inguinal area of FGF21KO host mice (KO→KO and WT→KO groups). WT Sham and FGF21KO Sham groups had surgery in the same area, but no fat was transplanted. The first transplantation cohort was used to confirm the survival of the graft tissue by hematoxylin and eosin staining and check blood supply of the graft by Evans blue dye injection into inferior vena cava at 2 weeks after the surgery. The second cohort was used for measurement of body weight and body composition, and glucose metabolism. Another two cohorts were used for hyperinsulinemic–euglycemic clamp studies and flow cytometry analysis. Mice were killed for collection of blood sample and various tissues for further biochemical and histological evaluations.

**Hyperinsulinemic–euglycemic clamp in mice**. Male FGF21KO and WT mice after rmFGF21 treatment or fat transplantation were studied by hyperinsulinemic–euglycemic clamp to assess GIR, endogenous glucose production and insulin-stimulated glucose uptake. Mice were catheterized in the right internal jugular vein 3–4 days prior to clamp experiment. After fasting for 6 h, [3-$^3$H] glucose (PerkinElmer) was infused at 0.05 μCi min$^{-1}$ for 2 h of the pretreatment period and blood samples were collected at the end. Then, the infusion of [3-$^3$H] glucose was continued for 2 h of clamp period at 0.1 μCi min$^{-1}$ together with a bonus infusion of 300 mU kg$^{-1}$ human insulin (Novolin R, Novo Nordisk) and then a continuous infusion of insulin at a rate of 6 mU kg$^{-1}$ min$^{-1}$ to raise plasma insulin within a physiological range. Plasma glucose concentration was measured at 10 min intervals, and 20% glucose was infused at variable rates to maintain plasma glucose at basal concentrations. Plasma glucose values were maintained stable during clamp experiments (Supplementary Fig. 9b). At 120 min of clamp period, a bonus of 10 μCi 2-deoxy-D-[1-$^{14}$C] glucose (2-[$^{14}$C]DG) (PerkinElmer) was administered. After 35 min, the mice were killed and tissues (SAT, VAT, muscle and BAT) were taken for glucose uptake analysis. Basal and insulin-stimulated endogenous glucose production was estimated with glucose concentration and [3-$^3$H]glucose level during the pretreatment and the clamp period. Individual glucose uptake for each tissue was estimated with 2-[$^{14}$C]DG concentration in plasma and tissue sample. This protocol has been adapted from established protocol[61] and our previous studies[62,63].

**Flow cytometry analysis of macrophages in adipose SVF**. The SVF was isolated from inguinal subcutaneous or epidimal visceral fat depot of male FGF21KO and WT mice after rmFGF21 treatment or fat transplantation. The mice were anesthetized and the fat pads were removed and digested in 0.1% (w/v) collagenase type I (Invitrogen) for 30 min at 37 °C with gentle shaking. The digestion mixture was passed through a 100 mm cell strainer (BD Biosciences) and centrifuged at 800×g for 10 min at 4 °C. The SVF pellets were collected and washed twice, 1 × 10$^5$ freshly isolated cells were triple stained with F4/80-PE (Abcam, ab105156, clone CI:A3-1, 1:50), cd206-Alexa Fluor 647 (Biolegend, 141712, clone C068C2, 1:100) and cd11c-FITC (BD Biosciences, BD 557400, clone HL3, 1:100) on ice for 30 min in dark. Species-matched IgG was used as non-specific isotype controls. After washing, the cells were analyzed with LSR Fortessa Cell Analyzer (BD Biosciences).

**Real-time PCR and immunoblot analysis**. Total RNA was extracted by Trizol (Invitrogen) and reverse transcribed into cDNA using the ImProm-II reverse transcriptase (Promega). Real-time PCR reactions were performed using Quantifast SYBR Green master mix (QIAGEN) on a Light Cycler 480 system (Roche), normalized with the mouse *Gapdh* or human 18s gene. The primer sequences are listed in Supplementary Table 2. Proteins were extracted from tissues in RIPA buffer (0.5% NP40, 0.1% sodium deoxycholate, 150 mM NaCl, 50 mM Tris HCl [pH 7.4]) containing complete protease inhibitor cocktail (Roche). PVDF membrane was

probed with primary antibodies human βklotho (R&D, AF5889-SP, goat polyclonal, 0.5 ug ml$^{-1}$), mouse βklotho (R&D, AF2619, goat polyclonal, 0.5 ug ml$^{-1}$), GAPDH (Cell Signaling Technology, 3662 s, rabbit polyclonal, 0.2 ug ml$^{-1}$), phospho-Akt (S473, AIS, HKU, rabbit polyclonal, 0.2 ug ml$^{-1}$), Akt (Cell Signaling Technology, 9272, rabbit polyclonal, 0.2 ug ml$^{-1}$) and UCP-1 (Abcam, 10983, rabbit polyclonal, 0.2 ug ml$^{-1}$). The protein bands were visualized with enhanced chemiluminescence reagents (GE Healthcare) and quantified using the NIH ImageJ software. The uncropped scans of the immunoblots are included in Supplementary Figs. 11–12. Concentrations of FGF21 and total adiponectin in serum of mice models were quantified with ELISA kits from AIS, HKU.

**Histological and immunohistochemistry analysis.** Adipose tissues were fixed overnight in buffered formaldehyde (10%) and embedded in paraffin and were prepared at the thickness of 5 mM. The sections were deparaffinized and dehydrated with three changes of xylene for 15 min each and two changes of 100% alcohol for 5 min each. Sections were stained with haematoxylin (Sigma-Aldrich) for 1 min and differentiated in 1% acid alcohol for 10 s before counter-stained in eosin–phloxine solution (Sigma-Aldrich) for 5–6 s. Sections were then cleared in three changes of xylene for 5 min each and mounted with histofluid mounting medium (Marienfeld-Superior, Germany). Cell size quantification was performed using software by calculating pixels of adipocytes on HE staining. For immunohistochemistry, sections were sequentially incubated with primary antibody UCP-1 (Abcam, 10983, rabbit polyclonal, 0.5 ug ml$^{-1}$) overnight and anti-rabbit secondary antibody (Cell Signaling Technology, 7074 s, 0.1 ug ml$^{-1}$) for 1 h at room temperature, followed by development with 3,3-diaminobenzidine solution (Sigma-Aldrich). The nuclei were counter-stained with haematoxylin. Two independent investigators blinded to sample identity performed the staining and analyzed the adipose tissue sections respectively.

**Primary adipocyte differentiation assays.** SVF preadipocytes were isolated from SAT and VAT of 6-week male WT, FGF21KO and Klb AdipoKO mice as in flow cytometry experiment. Cells were grown to confluency and differentiated in vitro over an 8-day period. Differentiation was induced with 2-day treatment of high glucose DMEM containing 10% FBS, 0.5 mM isobutylmethylxanthine, 1 μM dexamethasone, 2 μM rosiglitazone and 10 μg ml$^{-1}$ insulin, and with 6-day treatment of high glucose DMEM containing 10% FBS and 10 μg ml$^{-1}$ insulin. Media was replaced every other day. Cells were treated with 200 ng ml$^{-1}$ or 400 ng ml$^{-1}$ rmFGF21. Differentiated cells were collected for measuring related gene expression or Oil Red O staining. Stained cells were visualized with an Olympus biological microscope BX41, and images were captured with an Olympus DP72 color digital camera.

**2-Deoxyglucose (2-DG) uptake assay.** The differentiated SVF adipocytes were treated with IL10 and TNFα for 72 h. After that, glucose uptake was assessed using a 2-DG Uptake Assay Kit according to the manufacturer's instructions (Abcam, ab136955). Briefly, adipocytes were starved in serum free medium overnight and then Krebs–Ringer–Phosphate–Hepesbuffer with 2% BSA for 40 min. After insulin stimulation, the glucose analog 2-DG was added to cells and the accumulated 2-DG6P was oxidized to generate NADPH, which resulted in oxidation of a substrate. The oxidized substrate then can be detected at OD = 412 nm.

**Statistics.** All analyses were performed with SPSS 14.0. Data were expressed as mean ± SEM. Clinical data which were normally distributed were shown as mean ± SD and those were not normally distributed, as determined using Kolmogorox–Smirnov test, were logarithmically transformed before analysis and expressed as median with interquartile range. Animal sample size for each study was chosen on the basis of literature documentation of similar well-characterized experiments, and no statistical method was used to predetermine sample size. Statistical significance was calculated by Student's $t$ test, Pearson's correlations, one-way analysis of variance (ANOVA) or two-way ANOVA. When ANOVA indicated a significant difference among the groups, the statistical difference between two groups was compared using a stricter criterion for statistical significance according to the Bonferroni rule. We did not exclude any samples in animal experiments. Experiments were not blind, except for those that were specified. A two-sided P value of less than 0.05 was considered significant.

**Data availability.** The data that support the findings of this study are available from the corresponding authors upon reasonable request.

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

## Acknowledgements

We thank Prof. Sheng-Cai Lin (Xiamen University) for valuable comments on the manuscript. We also thank Prof. Ed X. Wu and Dr Anna M. Wang (The University of Hong Kong) for MRI measurement in mice. We also thank Dr Sheng Liu, Dr Yunxia Zhu (Shanghai Jiao Tong University Affiliated Sixth People's Hospital) and Mr. Siyuan Pan (Department of Computer Science and Engineering, Shanghai Jiao Tong University) for excellent technical support in data collection and image analysis. This work was supported by the National 973 project of China (2011CB504001), Natural Science Foundation of China (NSFC) major international (regional) joint research project (81220108006) and NSFC-NHMRC joint research grant (81561128016) to W.J., Hong Kong Research Grant Council (781413 M) to A.X., Hong Kong Scholars Program (XJ2013035), Young Scientists Fund of NSFC (81200292), Shanghai Pujiang Program (17PJ1407500) and Municipal Human Resources Development Program for Outstanding Young Talents in Medical and Health Sciences in Shanghai (2017YQ009) to H.L., and NSFC grant (61572316) to B.S.

## Author contributions

H.L. designed the study, carried out the research, analyzed, interpreted the results, and wrote the manuscript. G.W. designed the study, carried out the research, and analyzed the results. M.Z. contributed to the hyperinsulinaemic–euglycaemic clamp experiments. X.H., and L.W. contributed to the clinical research. B.S. conducted image-based 3D-reconstruction of abdominal MRI and automatic image segmentation of adipocytes. Q.F., P.L., Y.B. reviewed the manuscript. A.X. designed the study, wrote and edited the manuscript and W.J. designed the study and reviewed the manuscript.

## Additional information

**Competing interests:** The authors declare no competing financial interests.

