## [Peer Review File · Nature Communications]

Reviewers' Comments:

Reviewer #1 (Remarks to the Author)

FGF21 mostly plays roles as an endocrine signal modulating energy homeostasis. The pharmacological effects of FGF21 in obesity and diabetes have been widely recognized. In this manuscript, authors have examined physiological roles of FGF21 in systemic energy homeostasis in humans and mice, reporting findings described below.

- 1) Endogenous FGF21, of which serum levels are elevated in obesity, possibly serves as a defense mechanism against systemic insulin resistance in humans.
- 2) Specific expansion of subcutaneous fat, but not visceral fat, by FGF21 promotes systemic insulin sensitivity in mice, indicating a possible therapeutics against insulin resistance and type 2 diabetes by targeting subcutaneous fat through manipulation of FGF21

The results reported in this manuscript include potentially interesting findings. However, as there are some concerns described below, I cannot recommend this manuscript to be accepted for publication at present. Authors should clear these concerns.

Authors are required to response several points described below.

Major points

1. Line 34 and others: Authors should clear whether "biogenesis (expansion) of subcutaneous fat" results from hyperplasia or hypertrophy of adipocytes.
2. Lines 37-40 and others: Serum FGF21 are completely abolished liver-specific FGF21 knockout mice, indicating that serum FGF21 levels are primary derived from the liver. In addition, liver-specific FGF21 knockout mice with diet-induced obesity have increased insulin resistance, but adipose-specific FGF21 knockout mice have not (Markan KR et al., Diabetes (2014) 63, 4057-63). These results by Markan et al. are apparently inconsistent with those reported in this manuscript. It is important to clear this problem. As one way to resolve it, I suggest authors to reexamined them by using adipose-specific FGF21 knockout mice.
3. Lines 194-195 and others: As describe above, serum FGF21 levels are primary derived from the liver but not adipocytes (Markan KR et al., Diabetes (2014) 63, 4057-63). How did transplanted subcutaneous fat improve hepatic insulin sensitivity and glucose uptake in endogenous SAT of FGF21 KO mice?
4. The legend of supplemental Figure 4: Preadiocytes were treated with the pharmacological dose (400 ng/ml) of recombinant FGF21. As authors examined physiological roles of FGF21, they should treat them with the physiological dose of recombinant FGF21.

Minor points

1. Lines 56-57: FGF21 deficient mice indicated here are adenovirus-mediated hepatic FGF21knockdown mice. The FGF21 KO mice line used in the manuscript did not develop steatosis by ketogenic diet (Murata et al. PLoS One (2013) 8, e69330).
2. Lines 230-232: Was K1b efficiently removed in K1bAdipo KO differentiated adipocytes? Authors should show the results as supplemental data. In addition, what does "cell autonomus" mean in this manuscript?
3. Line 394: Authors should refer the paper (the ref. 35) that originally reported the FGF21 KO mice line instead of the ref. 44.
4. Lines 397-399: Although authors describe that adipose tissue-specific K1b knockout mice were confirmed by both genotyping and RT-qPCR analysis, they do not show the results. Authors should show the results as supplemental data.

Reviewer #2 (Remarks to the Author)

The study by Jia et al. reports a set of observations on the role of FGF21 targeting specifically the subcutaneous depot of WAT and by this means promoting a metabolically healthy response that includes enhanced insulin sensitivity.

Considering the overall statements of the manuscript pointing to SAT expansion as key for the healthy metabolic responsiveness to FGF21, it is surprising that the authors ignore throughout the manuscript the behavior of the BAT depot (with the exception of negative data for insulin sensitivity, thus suggesting that samples are likely to be available,..) and the extent of browning of SAT. The crude data for interscapular BAT weight changes and other functional parameters would be necessary to provide a comprehensive picture. As the authors state in the Introduction, brown and beige (i.e. SAT with infiltrated beige cells) adipose tissues are recognized targets of FGF21 action, and in fact a major effect of FGF21 on SAT is to promote the browning/beiging of the tissue (Fisher et al. Ref 12). Is this browning effect the underlying cause of the positive action of FGF21 on systemic metabolism via SAT "expansion"? Data on this issue is essential for a comprehensive understanding of the events taking place in the distinct rodent models used. It is really surprising that the large effort placed to characterize FGF21 action in SAT (including events such as M2 recruitment, known to be closely associated with browning in SAT, see Chawla papers in Cell in the last years) does not include any type of characterization of this phenomenon otherwise known to be highly relevant to confer metabolically healthy and insulin sensitive status to SAT. The lack of such characterization is a substantial weakness of the manuscript as it does preclude a comprehensive interpretation of multiple observations resulting from experiments.

Specific points

1. FGF21 levels in the IRO group should be provided (Fig2), as done before for the ISO group (Fig 1); are they the same or are higher than those in ISO patients? These data are needed for a comprehensive assessment of the rationale at the beginning of the manuscript.
2. Initial data on beta-Klotho expression, behaving differentially according to the WAT depot (in humans and in mice) should be contrasted with published literature, which is not mentioned. For humans, the reference to Gallego-Escuredo et al. paper in IJO 2015, reporting data for beta-Klotho in SAT and VAT adipose tissues from obese patients at distinct metabolic status should be discussed as they are considerably contradictory with findings in the current manuscript. Moreover, the closely related study in monkeys (Nygaard, IJO 2014) in SAT and VAT depots also showing distinct results on beta-Klotho expression from those in the manuscript, deserves to be commented.
3. Overall in the manuscript, the way in which data of relative weight of adipose depots are provided should be improved. Percentage data is poorly informative and representation of the actual mg or g data would be better to catch the actual relevance of quantitative changes. Moreover, the expression of the weight of depots relative to body weight is somewhat incorrect as the body weight includes the weight of the adipose depot to what it is referred. The data of adipose depots should be provided as crude weight and, much better, relative to the size of the animal regardless of the extent of adiposity, referral to tibia length for instance is a good choice.
4. Provision of data on how the status of beta-Klotho levels in several experimental models is lacking (e.g., does replacement with large amounts of recombinant FGF21 to FGF21-KO mice, leading to amelioration of insulin sensitivity, are causing changes in beta-Klotho expression?).
5. Do the cell autonomous effects found in cells cultures from SAT occur in cells coming from VAT? This type of experiment with VAT cells, as that reported in page 11, is required to sustain the SAT versus VAT distinction underlying the whole statement in the paper.
6. The apparently conclusive statement from the transplantation experiments using WT and FGF21-KO adipose pads regarding direct effects of FGF21 (first paragraph, page 16) should be toned down. The lack or presence of FGF21 in the transplanted fat may result in plenty of changes in adipokine release (e.g. adiponectin, as suggested by the experiments, or many others) and indirect effects cannot be ruled out. Otherwise, measurement of systemic FGF21 levels in FGF21-KO mice transplanted with WT adipose pads is required for the interpretation of the experiment.
7. Regarding the FGF21 replenishment study (the use of Alzet mini-pumps for 4-week delivery of FGF21 ; with no signs of degradation after 1 month, as stated in Fig3f), referenced as ref. 49 in methods, this reviewer couldn't find any mention of this technique for long-term delivery in

Reference 49. Please clarify.

Reviewer #3 (Remarks to the Author)

The manuscript "Fibroblast growth factor 21 increases insulin sensitivity through specific expansion of subcutaneous fat" by Li et al. presents data from human subjects and mouse models to propose a mechanism by which FGF21 improves insulin action. The authors start by showing that FGF21 levels are positively correlated with subcutaneous adipose tissue expansion in humans who are insulin sensitive and overweight (ISO). They then studied FGF21 knockout mice and adipose-specific Beta-klotho knockout mice and showed these animals have decreased subcutaneous fat mass and more insulin resistance when fed a high-fat diet. This phenotype could be reversed by treating with FGF21 or by transplanting subcutaneous fat from wild type donors to FGF21 knockout recipients. This manuscript provides further details on the role of FGF21 in adipose biology and metabolism, but does not represent a new conceptual advance that would be of broad interest to the readership of this journal. Moreover, this manuscript is lacking in a mechanistic explanation and the data do not fully justify the conclusions made.

Below are some specific points of criticism:

Major

1. There are already a number of quality manuscripts describing how FGF21 affects systemic metabolism (Camporez, *Endocrinology*, 2013) and subcutaneous adipose tissue in particular (Fisher, *Genes and Development*, 2012). As a result, this protein is viewed as a promising drug target (Talukdar, *Cell Metabolism*, 2016). This study does not offer any significant new insights into the mechanism by which FGF21 acts and seems to be more appropriate for a subspecialty journal.

2. When they replenish FGF21 in Figure 5, how do they know that the benefits are mediated via action on the subcutaneous fat as opposed to another tissue? The title of the paper states that FGF21 mediates its effects via the subcutaneous fat, but this has not been formally shown. One option would be to replace FGF21 in mice with a double FGF21 knockout who are also adipose-specific beta-klotho knockout. If the benefits of physiologic replacement of FGF21 were lost, this would help support the claim that FGF21 acts via subcutaneous fat.

Specific Points

1. The human data is interesting. However, it is not clear why the authors only describe the association between FGF21 levels and subcutaneous fat area in ISO and not in subjects who are insulin resistant and overweight/obese (IRO). They do study these individuals in Figure 2, so it seems odd that they were not part of the initial correlative analysis.

2. The blot in Figure 2D is cropped too tightly. Part of the immunoreactive band is cut-off in the figure.

3. When phenotyping the FGF21 knockout animals in Figure 3, the authors should measure lean mass as well as fat mass. Also, have they done any studies in female mice? Is this phenotype male specific or is it relevant in both genders?

4. Rather than just measuring fat mass in Figure 3, the authors should also do histologic analysis. Are there smaller fat cells?

5. For the insulin tolerance tests (Figure 4H, 5C, 6F), they should not report the data as a percentage of basal glucose, but should instead show the raw values. Moreover, a t-test is not the correct statistical test to analyze groups with repeated measures. An ANOVA is the more

appropriate test.

6. For the clamps, they should show that glucose values are actually maintained as intended in these studies.

7. In Figure 7, they argue that adipogenesis is impaired. However, they have only measured gene expression and not adipogenesis itself.

8. Were the studies in Figure 8 done on a high fat diet? This needs to be made clear.

Responses to Reviewer #1

FGF21 mostly plays roles as an endocrine signal modulating energy homeostasis. The pharmacological effects of FGF21 in obesity and diabetes have been widely recognized. In this manuscript, authors have examined physiological roles of FGF21 in systemic energy homeostasis in humans and mice, reporting findings described below.

1) Endogenous FGF21, of which serum levels are elevated in obesity, possibly serves as a defense mechanism against systemic insulin resistance in humans.

2) Specific expansion of subcutaneous fat, but not visceral fat, by FGF21 promotes systemic insulin sensitivity in mice, indicating a possible therapeutics against insulin resistance and type 2 diabetes by targeting subcutaneous fat through manipulation of FGF21

The results reported in this manuscript include potentially interesting findings. However, as there are some concerns described below, I cannot recommend this manuscript to be accepted for publication at present. Authors should clear these concerns.

Authors are required to response several points described below.

Major points

1. Line 34 and others: Authors should clear whether “biogenesis (expansion) of subcutaneous fat” results from hyperplasia or hypertrophy of adipocytes.

Answer: Thank you for your valuable suggestion. We totally agree that we should investigate whether the expansion of subcutaneous fat results from hyperplasia or hypertrophy of adipocytes. We have now checked the morphological changes and measured cell size distribution of subcutaneous adipose tissue after chronic treatment of FGF21. Histological analysis revealed that chronic treatment of rmFGF21 with a physiological dose increased number of small size adipocytes and decreased number of large adipocytes in subcutaneous fat of FGF21KO mice (Fig. 3i). Consistently, the mRNA levels of genes encoding proteins involved in adipogenesis (*cebpa*, *srebf1a*, *srebf1c*) were reduced in SAT of FGF21KO mice after HFD induction, and was partially restored after rmFGF21 treatment (Fig. 7a,c). These data suggest that the expansion of subcutaneous fat mainly results from hyperplasia of adipocytes. The increased small adipocytes and M2 macrophage polarization after FGF21 replenishment in FGF21KO mice (Fig. 3i, Fig. 8) suggest a relatively proper and healthy expansion of SAT (*Sun K, J Clin Invest, 2011; Strissel KJ, Diabetes, 2007*). These results were now described (line 166-172, 249-255) and discussed (line 352-354) in the manuscript.

2. Lines 37-40 and others: Serum FGF21 are completely abolished liver-specific FGF21 knockout mice, indicating that serum FGF21 levels are primary derived from the liver. In addition, liver-specific FGF21 knockout mice with diet-induced obesity have increased insulin resistance, but adipose-specific FGF21 knockout mice have not (Markan KR et al., Diabetes (2014) 63, 4057-63). These results by Markan et al. are apparently inconsistent with those reported in this manuscript. It is important to clear this problem. As one way to resolve it, I

suggest authors to reexamined them by using adipose-specific FGF21 knockout mice.

Answer: Thank you very much for your comment. We have reexamined our result in FGF21 adipose-specific knockout mice (FGF21 AdipoKO). Results showed that FGF21 AdipoKO mice did not have reduced circulating FGF21 level and increased insulin resistance after HFD induction. Body weight and fat distribution of FGF21 AdipoKO mice were similar to those of WT littermates (Data shown as below).

However, we believe that these results regarding FGF21 adipose-specific knockout mice are consistent with our conclusion. We found that elevated circulating FGF21 maintained insulin sensitivity via expanding SAT under diet-induced obesity, **but we did not claim that adipose secreted FGF21 played a key role in that process.** As discussed in line 350-359, we propose that circulating FGF21 acts in an endocrine manner on subcutaneous fat to promote the healthy expansion, which in turn produces other factors (such as adiponectin or good adipokines) to promote systemic insulin sensitivity.

Figure Legend. Adipose tissue specific FGF21-knockout mice (FGF21 AdipoKO) did not have decreased fat mass and glucose intolerance after HFD induction.

Eight-week-old, male WT and FGF21 AdipoKO mice were fed with HFD for 8 weeks. n=8/group. (a) Generation of FGF21 AdipoKO mice. (b) Genotyping of FGF21 AdipoKO mice by genomic PCR. (c) Serum FGF21 levels at fed state in WT and FGF21 AdipoKO mice during HFD induction were measured

by ELISA. (d-f) Body weight, fat mass and lean mass were measured at various time periods. (g) Adipose tissue mass (SAT, epiVAT, periVAT and BAT depots) was measured in WT and FGF21 AdipoKO mice fed with HFD. (h) Glucose tolerance test (GTT) and (i) Insulin tolerance test (ITT) performed in WT and FGF21 AdipoKO mice fed with HFD for 8 weeks. Data are presented as mean \pm SEM. Significance was determined by student's *t* test.

3. Lines 194-195 and others: As describe above, serum FGF21 levels are primary derived from the liver but not adipocytes (Markan KR et al., Diabetes (2014) 63, 4057-63). How did transplanted subcutaneous fat improve hepatic insulin sensitivity and glucose uptake in endogenous SAT of FGF21 KO mice?

Answer: Sorry for the confusion. We do not claim that the beneficial effects from transplantation of subcutaneous fat pad on systemic glucose metabolism are attributed to circulating FGF21 derived from adipocytes. Instead, we think that subcutaneous fat transplantation performed in FGF21KO mice results in changes in adipokines release and other indirect effects, which may contribute to the systemic insulin sensitivity and increase glucose uptake in peripheral tissues such as SAT. In our study, the expression of adiponectin in endogenous SAT and serum adiponectin level were partially restored in FGF21KO mice after transplantation of WT subcutaneous fat (Fig. 6n,o). Moreover, number of M2 macrophage and anti-inflammatory factors (such as IL10 as demonstrated in our study) were enhanced in endogenous SAT after transplantation of WT subcutaneous fat (Fig. 8d-f). Taken together, subcutaneous fat transplantation may exert its metabolic benefits in FGF21KO mice by promoting the release of adiponectin and altering macrophage polarization and the production of other factors from the fat tissue.

Indeed, FGF21 has been shown to promote adiponectin release in adipose tissue (*Lin Z, Cell Metab, 2013; Holland WL, Cell Metab, 2013*). Adiponectin was reported to enhance hepatic insulin sensitivity by increasing insulin receptor substrate-2 expression and promoting the activation of AMPK in liver (*Awazawa M, Cell Metab, 2011; Nawrocki AR, J Biol Chem, 2005*). *In vitro* study also found globular adiponectin can enhance glucose uptake in adipocytes via the activation of AMPK (*Wu X, Diabetes, 2003*). In adipose tissue, M2-polarized resident macrophages are able to partially protect adipocytes from inflammatory factors and may block M1 polarization, which contributes to the maintenance of insulin sensitivity in mild obesity (*Lumeng CN, J Clin Invest, 2007*). Those above-mentioned indirect effects are now discussed in the manuscript (line 394-413).

4. The legend of supplemental Figure 4: Preadipocytes were treated with the pharmacological dose (400 ng/ml) of recombinant FGF21. As authors examined physiological roles of FGF21, they should treat them with the physiological dose of recombinant FGF21.

Answer: We have now used a dose of 200ng/ml for recombinant FGF21 treatment in our experiments. We revealed a dose-dependent effect of recombinant FGF21 in promoting adipogenesis of SVF derived from SAT of WT and FGF21KO mice, but not Klb AdipoKO mice (Supplementary Fig. 5). In previous studies, pharmacological dose of FGF21 treatment *in vitro* were set at 100nM (2000ng/ml) (*Díaz-Delfín J, Endocrinology, 2012*) or 50nM (1000ng/ml) (*Kharitononkov A, J Clin Invest, 2005; Hondares E, Cell Metab, 2010*). Pharmacological dose of FGF21 treatment *in vivo* were set at 1mg/kg/day in obese mice (*Coskun T, Endocrinology, 2008; Xu J, Diabetes, 2009*). We tested and verified that 0.1 mg/kg/day of continuous rmFGF21 treatment can mimic a level equivalent to those occurring in diet-induced obesity (Fig. 3d), which is nearly one tenth of the *in vivo* pharmaceutical dose. Considering this, we use 200ng/ml, nearly one tenth of pharmacological dose *in vitro*, as the physiological dose of rmFGF21 to treat

preadipocytes.

Minor points

1. Lines 56-57: FGF21 deficient mice indicated here are adenovirus-mediated hepatic FGF21 knockdown mice. The FGF21 KO mice line used in the manuscript did not develop steatosis by ketogenic diet (Murata et al. PLoS One (2013) 8, e69330).

Answer: We have corrected the description in the manuscript.

2. Lines 230-232: Was Klb efficiently removed in KlbAdipo KO differentiated adipocytes? Authors should show the results as supplemental data. In addition, what does “cell autonomus” mean in this manuscript?

Answer: 1. We have now shown that β klotho was efficiently removed in Klb AdipoKO differentiated adipocytes in Supplementary Fig. 9c. 2. We are sorry we might misuse the term “cell autonomous” in the manuscript. What we want to express is that cells from different fat depots can have distinct molecular and physiological properties. The differential roles of subcutaneous and visceral fat in glucose homeostasis cannot simply be attributed to their different locations. The depot-specific characteristics are preserved after isolation and in vitro differentiation of these cells. We have revised the related wording in the manuscript.

3. Line 394: Authors should refer the paper (the ref. 35) that originally reported the FGF21 KO mice line instead of the ref. 44.

Answer: We now refer the paper that originally reported the FGF21 KO mice line when we mentioned the mice line for the first time in Method (Line 464).

4. Lines 397-399: Although authors describe that adipose tissue-specific Klb knockout mice were confirmed by both genotyping and RT-qPCR analysis, they do not show the results. Authors should show the results as supplemental data.

Answer: Thank you very much for pointing out this issue. We now show the results in Supplementary Fig. 9b,c.

Responses to Reviewer #2:

The study by Jia et al. reports a set of observations on the role of FGF21 targeting specifically the subcutaneous depot of WAT and by this means promoting a metabolically healthy response that includes enhanced insulin sensitivity.

Considering the overall statements of the manuscript pointing to SAT expansion as key for the healthy metabolic responsiveness to FGF21, it is surprising that the authors ignore throughout the manuscript the behavior of the BAT depot (with the exception of negative data for insulin sensitivity, thus suggesting that samples are likely to be available,..) and the extent of browning of SAT. The crude data for interscapular BAT weight changes and other functional parameters would be necessary to provide a comprehensive picture. As the authors state in the Introduction, brown and beige (i.e. SAT with infiltrated beige cells) adipose tissues are recognized targets of FGF21 action, and in fact a major effect of FGF21 on SAT is to promote the browning/beiging of the tissue (Fisher et al. Ref 12). Is this browning effect the underlying cause of the positive action of FGF21 on systemic metabolism via SAT “expansion”? Data on this issue is essential for a comprehensive understanding of the events taking place in the distinct rodent models used. It is really surprising that the large effort placed to characterize FGF21 action in SAT (including events such as M2 recruitment, known to be closely associated with browning in SAT, see Chawla papers in Cell in the last years)

does not include any type of characterization of this phenomenon otherwise known to be highly relevant to confer metabolically healthy and insulin sensitive status to SAT. The lack of such characterization is a substantial weakness of the manuscript as it does preclude a comprehensive interpretation of multiple observations resulting from experiments.

Answer: Thank you for your excellent suggestion. We totally agree the data regarding browning of SAT and the behavior of BAT is important for a more comprehensive interpretation of our observations. These data have already been collected during our previous experiments. We have now incorporated the data into the manuscript.

As shown in Fig. 3h, the crude weight of BAT in FGF21KO mice did not have a significant change after the replenishment with rmFGF21 to a physiologically-relevant HFD-induced level. Results also showed that the levels of UCP-1 in SAT were not detectable after replenishing physiological dose of rmFGF21. HE and UCP-1 immunohistochemistry staining showed no sign of browning effect in SAT. In BAT, UCP-1 protein levels were similar among the three groups (Supplementary Fig. 4).

Fisher et al. found that while FGF21 knockout mice display impairment in cold environment-induced browning of SAT, pharmacologic treatment of FGF21 (24ug/day) only causes a very modest induction of thermogenic gene expression in WAT under ambient temperature (*Fisher FM, Genes Dev, 2012*), suggesting that FGF21 alone is not sufficient to drive full browning program. Our results suggest that the effects of physiological level of FGF21 (0.1mg/kg/day \approx 3ug/day, much less than the pharmacologic dose) in mediating glucose metabolism is contributed by the expansion of subcutaneous fat, which is independent of browning of WAT and the activation of BAT.

Besides browning, subcutaneous fat has been demonstrated to combat insulin resistance via multiple mechanisms. Compared with visceral fat, subcutaneous fat has more beneficial effects on systemic metabolism as shown by transplantation (*Tran TT, Cell Metab, 2008*). The paper suggests that subcutaneous fat is intrinsically different from visceral fat and produces substances that can act systemically to improve glucose metabolism. Several studies have demonstrated that subcutaneous fat can secrete more adiponectin and leptin than visceral fat. It has been reported that adiponectin and leptin genes were expressed at 10,000 to a million-fold higher in peripheral fat depots as compared to visceral fat depot in mice (*Satoor SN, Sci Rep, 2011*). In human, total adiponectin as well as leptin were higher in subjects with more subcutaneous fat than those with more visceral fat accumulation (*Taksali SE, Diabetes, 2008; Yatagai T, Metabolism, 2003*). Moreover, compared to visceral fat, subcutaneous fat secretes less inflammatory cytokines. The release of IL-6 was 2–3 times lower in subcutaneous adipose tissue than that in omental adipose tissue (*Fried SK, J Clin Endo Metab, 1998*). It was also found that visceral but not subcutaneous fat was associated with increasing levels of C-reactive protein (CRP) and TNF α (*Hocking S, Endocrine Rev, 2013*). Although the reason why subcutaneous fat is less prone to develop inflammation is unclear yet, one possible explanation is due to the protein inhibitor of activated STAT 1 (PIAS1) which is highly expressed in subcutaneous fat depot. It was found that PIAS1 suppressed the inflammation in subcutaneous fat via inactivation of c-Jun N-terminal kinase (JNK) (*Liu Y, Diabetes, 2015; Shimizu, Diabetes, 2015*). These mechanisms might provide explanations to the benefits of expansion of subcutaneous fat caused by physiologically-relevant HFD-induced FGF21 although browning of SAT is not affected. We have discussed this issue in the revised version (Line 377-393).

Specific points

1. FGF21 levels in the IRO group should be provided (Fig2), as done before for the ISO group (Fig 1); are they the same or are higher than those in ISO patients? These data are needed for a comprehensive assessment of the rationale at the beginning of the manuscript.

Answer: Based on your suggestion, we have now included 30 IRO subjects for the analysis of circulating FGF21 levels. The data is included in Fig. 1. Although the total fat mass was similar in the recruited subjects with ISO and IRO, fat distribution differed remarkably in these two groups. Subcutaneous fat area (SFA) in subjects with ISO was significantly higher than that in subjects with IRO, whereas visceral fat area (VFA) in subjects with ISO was significantly lower than that in subjects with IRO. SFA to VFA ratio in ISO was significantly higher than that in NW, while the ratio in IRO was lower than that in NW (Fig. 1a-d). The result of hyperinsulinemic-euglycemic clamp confirmed that unlike subjects with IRO, ISO subjects did not have an obvious reduction in glucose infusion rate (GIR) (Fig. 1e), suggesting that the increased fat mass which was mainly displayed in subcutaneous region led to better insulin sensitivity. Serum adiponectin level significantly decreased in subjects with IRO but remained unchanged in subjects with ISO. Notably, although FGF21 level in both ISO and IRO groups were higher than those in normal weight subjects, serum FGF21 levels were markedly higher in subjects with ISO (194.66pg/ml [119.66, 256.86]) than subjects with IRO (134.27pg/ml [88.95, 207.78]) ($P<0.05$) (Fig. 1f). Furthermore, SFA were positively correlated with serum FGF21 levels in subjects with ISO ($r=0.450$, $P<0.05$) (Fig. 1g). However, no significant relationship between VFA and FGF21 was found in these subjects (Fig. 1h). Serum FGF21 was independently associated with SFA after the adjustment for serum adiponectin level ($P<0.001$). These clinical findings suggested that increased serum FGF21 in ISO were closely correlated with the increased subcutaneous fat which may contribute to the maintenance of insulin sensitivity. The description and discussion of the results were included in the manuscript (line 78-122, 317-333).

2. Initial data on beta-Klotho expression, behaving differentially according to the WAT depot (in humans and in mice) should be contrasted with published literature, which is not mentioned. For humans, the reference to Gallego-Escuredo et al. paper in IJO 2015, reporting data for beta-Klotho in SAT and VAT adipose tissues from obese patients at distinct metabolic status should be discussed as they are considerably contradictory with findings in the current manuscript. Moreover, the closely related study in monkeys (Nygaard, IJO 2014) in SAT and VAT depots also showing distinct results on beta-Klotho expression from those in the manuscript, deserves to be commented.

Answer: Thank you very much for your advice. We have now discussed our results and published literature regarding β klotho expression thoroughly in the manuscript. In the paper in IJO 2015 by Gallego-Escuredo et al., β klotho level was significantly decreased in subcutaneous fat in obese subjects with normal glucose level. The seemingly contradictory findings with our study is that we found β klotho in subcutaneous fat was elevated in ISO and decreased in IRO. However, in Gallego-Escuredo paper, the BMI of obese subjects with normal glucose level was 39.2 ± 1.2 and their average HOMA-IR was 3.3 ± 0.3 , so these subjects are insulin-resistant and belong to IRO group in our study. Thus, the result in Gallego-Escuredo paper is consistent with that of our study. In Nygaard, IJO 2014, HFD-resistant monkeys were defined as animals chronically maintained on HFD that have body weights and glucose stimulated insulin secretion (GSIS) within 2 standard deviations of the controls while HFD-sensitive monkeys had significant increased body weight, fat mass, GSIS, as well as decreased glucose clearance during an i.v.GTT. Compared to HFD-sensitive monkeys, HFD-resistant monkeys had increased β klotho level in subcutaneous fat, which are consistent with our results. The above discussion has been included in the manuscript (line 326-333).

3. Overall in the manuscript, the way in which data of relative weight of adipose depots are

provided should be improved. Percentage data is poorly informative and representation of the actual mg or g data would be better to catch the actual relevance of quantitative changes. Moreover, the expression of the weight of depots relative to body weight is somewhat incorrect as the body weight includes the weight of the adipose depot to what it is referred. The data of adipose depots should be provided as crude weight and, much better, relative to the size of the animal regardless of the extent of adiposity, referral to tibia length for instance is a good choice.

Answer: Based on your suggestion, we now use crude weight of fat depots in Fig. 3c, 3h and 4f. The conclusion remains unchanged. We will record the tibia length in the future experiments.

4. Provision of data on how the status of beta-Klotho levels in several experimental models is lacking (e.g., does replacement with large amounts of recombinant FGF21 to FGF21-KO mice, leading to amelioration of insulin sensitivity, are causing changes in beta-Klotho expression?).

Answer: We now examined the expression level of β klotho after FGF21 treatment. Treatment with rmFGF21 does not change β klotho expressions in SAT and epiVAT (Supplemental Fig. 3f,g).

5. Do the cell autonomous effects found in cells cultures from SAT occur in cells coming from VAT? This type of experiment with VAT cells, as that reported in page 11, is required to sustain the SAT versus VAT distinction underlying the whole statement in the paper.

Answer: We examined preadipocytes derived from epiVAT of WT, FGF21KO and FGF21KO+rmFGF21 mice. Results showed that treatment with rmFGF21 had minimal effect or almost no effect on the adipogenesis and lipid accumulation of primary adipocytes from VAT. Now it has been described in the manuscript (line 268-271).

6. The apparently conclusive statement from the transplantation experiments using WT and FGF21-KO adipose pads regarding direct effects of FGF21 (first paragraph, page 16) should be toned down. The lack or presence of FGF21 in the transplanted fat may result in plenty of changes in adipokine release (e.g. adiponectin, as suggested by the experiments, or many others) and indirect effects cannot be ruled out. Otherwise, measurement of systemic FGF21 levels in FGF21-KO mice transplanted with WT adipose pads is required for the interpretation of the experiment.

Answer: Thank you for pointing out the issue. We have included discussion regarding indirect effects of subcutaneous fat transplantation and toned down the conclusive statement in the manuscript (Line 367-376). Subcutaneous fat transplantation may exert its metabolic benefits in FGF21KO mice by increasing the release of adiponectin and promoting M2 macrophage polarization and its related cytokine such as IL10 as demonstrated in our study.

Indeed, FGF21 has been shown to promote adiponectin release in adipose tissue (*Lin Z, Cell Metab, 2013; Holland WL, Cell Metab, 2013*). Adiponectin was reported to enhance hepatic insulin sensitivity by increasing insulin receptor substrate-2 expression and promoting the activation of AMPK in liver (*Awazawa M, Cell Metab, 2011; Nawrocki AR, JBC, 2005*). *In vitro* study also found globular adiponectin can enhance glucose uptake in adipocytes via the activation of AMPK (*Wu X, Diabetes, 2003*). In adipose tissue, M2-polarized resident macrophages are able to partially protect adipocytes from inflammatory factors and may block M1 polarization, which contributes to the maintenance of insulin sensitivity in mild obesity (*Lumeng CN, J Clin Invest, 2007*). Those above-mentioned indirect effects are now discussed in the manuscript (line 394-413).

7. Regarding the FGF21 replenishment study (the use of Alzet mini-pumps for 4-week delivery

of FGF21 ;with no signs of degradation after 1 month, as stated in Fig3f), referenced as ref. 49 in methods, this reviewer couldn't find any mention of this technique for long-term delivery in Reference 49. Please clarify.

Answer: Thank you for pointing out this issue. In ref. 49 in last version (Xu A, *J Clin Invest*, 2003), this long-term delivery technique was used for 2-week delivery of adiponectin. This citation is now replaced with another paper from our lab (Lin Z, *Cell Metab*, 2013), in which this technique was used for 4-week delivery of recombinant FGF21. The Alzet osmotic system is widely used for chronic delivery of proteins in a constant rate (http://www.alzet.com/products/guide_to_use/proteins_and_peptides.html). We found that recombinant FGF21 in the pump is rather stable, and there is no degradation after 4 weeks.

Reviewer #3 (Remarks to the Author):

The manuscript “Fibroblast growth factor 21 increases insulin sensitivity through specific expansion of subcutaneous fat” by Li et al. presents data from human subjects and mouse models to propose a mechanism by which FGF21 improves insulin action. The authors start by showing that FGF21 levels are positively correlated with subcutaneous adipose tissue expansion in humans who are insulin sensitive and overweight (ISO). They then studied FGF21 knockout mice and adipose-specific Beta-klotho knockout mice and showed these animals have decreased subcutaneous fat mass and more insulin resistance when fed a high-fat diet. This phenotype could be reversed by treating with FGF21 or by transplanting subcutaneous fat from wild type donors to FGF21 knockout recipients. This manuscript provides further details on the role of FGF21 in adipose biology and metabolism, but does not represent a new conceptual advance that would be of broad interest to the readership of this journal. Moreover, this manuscript is lacking in a mechanistic explanation and the data do not fully justify the conclusions made.

Below are some specific points of criticism:

Major

1. There are already a number of quality manuscripts describing how FGF21 affects systemic metabolism (Camporez, *Endocrinology*, 2013) and subcutaneous adipose tissue in particular (Fisher, *Genes and Development*, 2012). As a result, this protein is viewed as a promising drug target (Talukdar, *Cell Metabolism*, 2016). This study does not offer any significant new insights into the mechanism by which FGF21 acts and seems to be more appropriate for a subspecialty journal.

Answer: Thank you very much for your comment. Maybe we did not give a clear description about the conceptual novelty of our study. FGF21 has recently attracted great attention due to its multiple therapeutic potentials for obesity-related metabolic complications. However, an unsolved puzzle is that “despite its pleiotropic metabolic benefits, FGF21 is paradoxically elevated in obesity and diabetes in both animals and humans” (Zhang X, *Diabetes*, 2008; Chen C, *Diabetes Care* 2011; Dushay J, *Gastroenterology*, 2010). Although early studies suggest that elevated FGF21 in obesity may reflect the existence of FGF21 resistance, the subsequent study did not support such a conclusion (Hale C, *Endocrinology*, 2012). This long-standing puzzle is solved by our study which finds out the pathophysiological role of elevated circulating FGF21 in obesity as a defense mechanism. We believe that there are several conceptual advances in this study, which are summarized as below:

1 Elevated endogenous FGF21 in obesity serves as a defense mechanism against systemic insulin resistance. Although the pharmacological effects of FGF21 have been well documented in animals and humans, its pathophysiological relevance remains **poorly understood**. In particular, it remains **unclear** as to how endogenous FGF21 exerts its systemic functions as a stress hormone in response to excess energy intake. We provide both animal and clinical evidences demonstrating for the **first time** that **elevated endogenous FGF21 in obesity serves as a defense mechanism against systemic insulin resistance.**

2 We discovered for the first time that FGF21 exerts its actions in an adipose depot-specific manner and causes specific expansion of subcutaneous fat, which is a novel mechanism to combat insulin resistance and metabolic dysregulation. Adipose tissue is a highly heterogeneous endocrine organ. Individuals with central obesity are more susceptible to developing diabetes and cardiovascular complications, whereas those with peripheral obesity are more metabolically healthy (Abate N, *J Clin Invest*, 1995; McLaughlin T, *J Clin Endo Metab*, 2011; Kwok KH, *Exp Mol Med.*, 2016). Further, a recent study from a European population showed that increased femoral subcutaneous fat mass is protective of cardiometabolic diseases and mortality (Stefan N, *Cell Metab*, 2017). However, the regulators that specifically control subcutaneous fat expansion remain poorly understood. Furthermore, although adipose tissue has been well documented as the primary target of FGF21, the fat depot-specific role of FGF21 has never been reported before. In this regard, our current study provides a series of animal and clinical evidences demonstrating that **FGF21 specifically promoting biogenesis of subcutaneous fat, but not visceral fat in diet-induced obesity. Such an adipose-specific action of FGF21 is attributed to high level of expression of the FGF21 receptor complex. Although a previous study shows that FGF21 acts in subcutaneous fat to promote browning under cold environment, whether such an effect contributes to improvement in systemic insulin sensitivity has not been reported.**

3 Our work provides novel mechanistic insights how subcutaneous fat acts as an endocrine tissue to alleviate systemic insulin resistance. Although clinical studies generally support a beneficial role of subcutaneous depots in the overall cardiometabolism, the molecular basis remains **poorly understood**. We demonstrate for the **first time** that transplantation of subcutaneous fat from wild-type to FGF21 knockout mice improved insulin sensitivity. FGF21 may exert its metabolic benefits in subcutaneous fat by the production of adipokines (eg. adiponectin) and altering macrophage polarization. Therefore, this study provides novel explanations of how subcutaneous fat acts as a beneficial tissue to metabolic health.

4. To the best of our knowledge, our study is the first to identify a hormonal factor that regulate the selective expansion of a specific adipose depot. Therefore, our study will provide a powerful tool for studying differential regulation and function of different adipose depots.

2. When they replenish FGF21 in Figure 5, how do they know that the benefits are mediated

via action on the subcutaneous fat as opposed to another tissue? The title of the paper states that FGF21 mediates its effects via the subcutaneous fat, but this has not been formally shown. One option would be to replace FGF21 in mice with a double FGF21 knockout who are also adipose-specific beta-klotho knockout. If the benefits of physiologic replacement of FGF21 were lost, this would help support the claim that FGF21 acts via subcutaneous fat.

Answer: Our conclusion that FGF21 exerts its metabolic benefits by expansion of subcutaneous fat was supported by a series of clinical, in vivo and in vitro evidences. First, in insulin-sensitive obese (ISO) individuals, we observed a close correlation among serum FGF21, amount of subcutaneous fat, and insulin sensitivity (Fig. 1). Second, high fat diet (HFD)-induced insulin resistance in FGF21 knockout mice is accompanied by selective reduction in subcutaneous fat, whereas these changes can be reversed by replenishment with physiological concentration of FGF21. Third, the insulin resistant phenotype of FGF21 knockout mice can be reversed by transplantation of subcutaneous fat from WT mice (Fig. 6). Fourth, recombinant FGF21 potently induces differentiation of subcutaneous adipocytes, but not visceral adipocytes (Supplementary Fig. 5).

According to the suggestion, we further investigated the effect of FGF21 in adipocyte-specific β klotho-knockout mice (Klb AdipoKO). The mice exhibited modestly decreased subcutaneous together with glucose intolerance after HFD induction (Fig. 4). After 8-week of HFD induction, we used physiological dose of rmFGF21 by osmotic pump (0.05mg/kg/day) to mimic HFD-induced circulating FGF21 level in KlbAdipoKO mice for another 4 weeks. The circulating FGF21 level was similar as the level during rmFGF21 replenishment in FGF21KO (Fig. 4e). However, unlike FGF21 KO mice, Klb AdipoKO are refractory to the effects of physiological concentrations of FGF21 on expansion of subcutaneous fat and alleviation of insulin resistance. Taken together, this new set of data further supports our conclusion that the metabolic benefits of FGF21 are mediated at least in part by expansion of subcutaneous fat. These results were now described and discussed in the manuscript (line 173-182, 190-193).

Specific Points

1. The human data is interesting. However, it is not clear why the authors only describe the association between FGF21 levels and subcutaneous fat area in ISO and not in subjects who are insulin resistant and overweight/obese (IRO). They do study these individuals in Figure 2, so it seems odd that they were not part of the initial correlative analysis.

Answer: Based on your suggestion, we have now included 30 IRO subjects for the analysis of circulating FGF21 levels. The data is included in Fig. 1. Although the total fat mass was similar in the recruited subjects with ISO and IRO, fat distribution differed remarkably in these two groups. Subcutaneous fat area (SFA) in subjects with ISO was significantly higher than that in subjects with IRO, whereas visceral fat area (VFA) in subjects with ISO was significantly lower than that in subjects with IRO. SFA to VFA ratio in ISO was significantly higher than that in NW, while the ratio in IRO was lower than that in NW (Fig. 1a-d). The result of hyperinsulinemic-euglycemic clamp confirmed that unlike subjects with IRO, ISO subjects did not have an obvious reduction in glucose infusion rate (GIR) (Fig. 1e), suggesting that the increased fat mass which was mainly displayed in subcutaneous region led to better insulin sensitivity. Serum adiponectin level significantly decreased in subjects with IRO but remained unchanged in subjects with ISO. Notably, although FGF21 level in both ISO and IRO groups were higher than those in normal weight subjects, serum FGF21 levels were markedly higher in subjects with ISO (194.66pg/ml [119.66, 256.86]) than subjects with IRO (134.27pg/ml [88.95, 207.78]) ($P<0.05$) (Fig. 1f). Furthermore, SFA were positively correlated with serum FGF21 levels in subjects with ISO ($r=0.450$, $P<0.05$) (Fig. 1g). However, no significant relationship between VFA and FGF21 was

found in these subjects (Fig. 1h). Serum FGF21 was independently associated with SFA after the adjustment for serum adiponectin level ($P < 0.001$). These clinical findings suggested that increased FGF21 in ISO were closely correlated with the increased subcutaneous fat which may contribute to the maintenance of insulin sensitivity. The description and discussion of the results were included in the manuscript (line 78-122, 317-333).

2. The blot in Figure 2D is cropped too tightly. Part of the immunoreactive band is cut-off in the figure.

Answer: We have now shown a larger immunoblots in Fig. 2d. All our original blots will be posted as a supplementary figures if our manuscript is accepted for publication.

3. When phenotyping the FGF21 knockout animals in Figure 3, the authors should measure lean mass as well as fat mass. Also, have they done any studies in female mice? Is this phenotype male specific or is it relevant in both genders?

Answer: Lean mass was measured and shown in Supplementary Fig. 2a. There was no significant difference of lean mass between FGF21KO and WT mice.

In female mice fed with HFD for 8 weeks, we also observed a similar trend although the insulin resistant phenotype in female FGF21 KO mice is less severe than the male mice.

4. Rather than just measuring fat mass in Figure 3, the authors should also do histologic analysis. Are there smaller fat cells?

Answer: As also suggested by Reviewer #1, we now investigated whether the expansion of subcutaneous fat results from hyperplasia or hypertrophy. We checked the morphological change and measured cell size distribution of subcutaneous adipose tissue after chronic treatment of FGF21. Histological analysis revealed that chronic treatment of rmFGF21 with a physiological dose increased number of small size adipocytes and decreased number of large adipocytes in subcutaneous fat of FGF21KO mice (Fig. 3i). Consistently, the mRNA levels of genes encoding proteins involved in adipogenesis (*cebpa*, *srebf1a*, *srebf1c*) were reduced in SAT of FGF21KO mice after HFD induction, and was partially restored after rmFGF21 treatment (Fig. 7a,c). These data suggest that the expansion of subcutaneous fat mainly results from hyperplasia of adipocytes. The increased small adipocytes and M2 macrophage polarization after FGF21 replenishment in FGF21KO mice (Fig. 3i, Fig. 8) suggest a relatively proper and healthy expansion of SAT (*Sun K, J Clin Invest, 2011; Strissel KJ, Diabetes, 2007*). These results were now described (line 166-172, 249-255) and discussed (line 352-354) in the manuscript.

5. For the insulin tolerance tests (Figure 4H, 5C, 6F), they should not report the data as a percentage of basal glucose, but should instead show the raw values. Moreover, a t-test is not the correct statistical test to analyze groups with repeated measures. An ANOVA is the more appropriate test.

Answer: We have reported the data in raw values in Fig. 4i, 5c, 6f. Repeated measures among groups are now analyzed using ANOVA as described in the legends.

6. For the clamps, they should show that glucose values are actually maintained as intended in these studies.

Answer: The figures of average glucose values during clamp experiments in humans and mice are now shown in Supplementary Fig. 8. The glucose levels were maintained stably during clamp.

7. In Figure 7, they argue that adipogenesis is impaired. However, they have only measured gene expression and not adipogenesis itself.

Answer: We showed the adipogenesis data in Supplemental Fig. 5c. Stromal vascular fraction (SVF) preadipocytes were isolated from SAT of WT and Klb AdipoKO mice. These preadipocytes were differentiated in an 8-day period in either the presence or absence of rmFGF21. Adipogenesis itself was observed and measured by oil red O staining. Consistent with the in vivo data, we found that recombinant FGF21 potently induces differentiation of subcutaneous adipocytes.

8. Were the studies in Figure 8 done on a high fat diet? This needs to be made clear.

Answer: Yes, experiments in Figure 8 were done on a high fat diet. We now make it clear in figure legend: “(a-c) After 8-week HFD induction and 4 weeks of treatment by vehicle or physiologically-relevant dose of rmFGF21 (0.1mg/kg/day), another cohort of WT+Vehicle, FGF21KO+Vehicle and FGF21KO+rmFGF21 mice were sacrificed for flow cytometry and other following experiments.” (Fig. 8 legend)

Reviewers' Comments:

Reviewer #1:

Remarks to the Author:

I have carefully examined responses by authors to my previous comments.

Their responses are adequate and the manuscript are adequately revised. I expect that the revised manuscript are now acceptable.

Reviewer #2:

Remarks to the Author:

The revised version of the manuscript by Jia et al. has incorporated new data and improved several points relative to the first version. Most of the points raised in the previous revision have been replied satisfactorily. Some other, however, remain raising some concerns.

- Regarding the general point of the BAT and SAT browning phenotyping of mouse models, some aspects remain unclear. Regarding BAT, a more precise assessment is required. iBAT size in FGF21-KO mice show systematically the same alteration (strongly significant reduction) than SAT (Fig 3c, 3 d). In the absence of gross characterization of BAT composition (which can be largely affected by relative amounts of fat versus protein), the lack of reversal of size due to FGF21 is not a data strong enough to rule out further characterization. Unchanged relative levels of UCP1 protein in the absence of the calculations of total UCP1 levels per iBAT depot are also not enough informative of BAT thermogenic capacity (as quoted by Nedergaard, & Cannon BBA, 2013 1831,943).

- Regarding SAT, the reasoning of the authors in their reply pointing to no role of browning of SAT may be right, but the data in Suppl. Fig 1 is rather surprising. A totally blank, "zero" signal for the UCP1 protein immunoblot in SAT in basal conditions (mice at 21°C, not thermoneutral, some extent of browning and UCP1 expression is expected to be there,...), is rather strange. There are multiple reports in the literature of some UCP1 in SAT in mice at room temperature. Perhaps common immunoblotting of BAT and SAT samples is not appropriate (due to the high UCP1 signal in BAT and corresponding short exposure of the blot to get a reasonable signal) is the reason but, in any case, it does not allow to appreciate if something happens with UCP1 in SAT.

- The requested data of FGF21 levels when FGF21-KO mice are transplanted with WT SAT is not provided, I remain thinking it is important to have a comprehensive picture of the model.

- The experimental basis of the statement that recombinant FGF21 is stable, with no degradation, during 4 weeks in the mini-pump (last Answer) is not provided. Was it checked in the referenced previous paper Lin Z et al.?

- The involvement of PPARgamma pathway should be toned down relative to the conclusive statements (including the last picture). The panel of transcripts depicted as "PPARgamma-targets" are not full standard and, in any case, a further experimentation beyond this type of very indirect approach (the genes tested have plenty of other transcriptional regulators) is needed for a full-blown claim for PPARgamma role.

Reviewer #3:

Remarks to the Author:

The revised manuscript by Li et al. entitled "Fibroblast growth factor 21 increases insulin sensitivity through specific expansion of subcutaneous fat" has been substantially strengthened by several additional experiments suggested by the reviewers.

All of the points made in my initial review have now been appropriately addressed by the authors. First, the authors now provide a clear rationale for the novel conceptual advance made by this paper. They explain that this paper is the first to show that FGF21 at physiological doses (as

opposed to pharmacological doses in many studies) appears to improve insulin sensitivity via hyperplasia of subcutaneous white fat. Although they do not directly measure hyperplasia, which can be done via BrdU incorporation, this is a reasonable inference to make from their morphological characterization and in vitro studies. Second, they have now generated adipocyte-specific beta-klotho knockout mice to show genetically that the benefits of physiological FGF21 are mediated via action on adipocytes. They have also satisfactorily addressed all of the minor points raised in my review.

I also want to point out, in reading the response to the other reviewers, that I think the authors have now shown that the actions of physiological FGF21 are unlikely to be mediated via effects on brown fat or beige fat biogenesis. Their new data includes UCP1 Western blots and immunostaining as shown in supplementary Figure 4. If they wanted to be even more definitive, this data could be further supported by measuring RNA levels of genes involved in the thermogenic pathway in subcutaneous fat and brown fat. Moreover, they could do tissue respiration assays to show that physiological FGF21 does not lead to increased O₂ consumption in subcutaneous fat or brown fat.

Overall, this is now an interesting manuscript that represents a nice advance in molecular metabolism. My only other suggestion would be to have the paper thoroughly edited for grammar and style to make it more straightforward to read.

Responses to Reviewer #1:

I have carefully examined responses by authors to my previous comments.

Their responses are adequate and the manuscript are adequately revised. I expect that the revised manuscript are now acceptable.

Responses to Reviewer #2:

The revised version of the manuscript by Jia et al. has incorporated new data and improved several points relative to the first version. Most of the points raised in the previous revision have been replied satisfactorily. Some other, however, remain raising some concerns.

- Regarding the general point of the BAT and SAT browning phenotyping of mouse models, some aspects remain unclear. Regarding BAT, a more precise assessment is required. iBAT size in FGF21-KO mice show systematically the same alteration (strongly significant reduction) than SAT (Fig 3c, 3d). In the absence of gross characterization of BAT composition (which can be largely affected by relative amounts of fat versus protein), the lack of reversal of size due to FGF21 is not a data strong enough to rule out further characterization. Unchanged relative levels of UCP1 protein in the absence of the calculations of total UCP1 levels per iBAT depot are also not enough informative of BAT thermogenic capacity (as quoted by Nedergaard, & Cannon BBA, 2013 1831,943).

Answer: Thank you for your suggestion. According to the reference²⁹ (Nedergaard & Cannon BBA, 2013), we agree that total UCP-1 protein level per mouse is the most physiologically relevant parameter for thermogenic capacity. We have now calculated total UCP-1 levels per iBAT depot by multiplying the UCP-1 protein level per mg homogenate protein with the total amount of proteins in each iBAT depot. No change was found in total UCP1 protein levels per mouse among the three groups including WT+Vehicle, FGF21KO+Vehicle and FGF21KO+rmFGF21 (Supplementary Figure 4b). These results are now described (Line 209-212) in the manuscript.

- Regarding SAT, the reasoning of the authors in their reply pointing to no role of browning of SAT may be right, but the data in Suppl. Fig 1 is rather surprising. A totally blank, "zero" signal for the UCP1 protein immunoblot in SAT in basal conditions (mice at 21°C, not thermoneutral, some extent of browning and UCP1 expression is expected to be there,...), is rather strange. There are multiple reports in the literature of some UCP1 in SAT in mice at room temperature. Perhaps common immunoblotting of BAT and SAT samples is not appropriate (due to the high UCP1 signal in BAT and corresponding short exposure of the blot to get a reasonable signal) is the reason but, in any case, it does not allow to appreciate if something happens with UCP1 in SAT.

Answer: Thank you for pointing out this issue. As you mentioned, signals of UCP1 protein

immunoblot might vary with experimental details, including loaded protein amount and exposure time in western blot. We now repeat UCP1 immunoblotting separately in SAT and BAT. We increased the loaded protein amount of SAT. The signal of UCP1 was detectable and still similar among three groups including WT+Vehicle, FGF21KO+Vehicle and FGF21KO+rmFGF21 (Supplementary Figure 4d). These results are now described (Line 212-213) in the manuscript.

- The requested data of FGF21 levels when FGF21-KO mice are transplanted with WT SAT is not provided, I remain thinking it is important to have a comprehensive picture of the model.

Answer: Thank you for your comment. We measured serum FGF21 levels in FGF21KO mice transplanted with WT SAT using ELISA. However, serum FGF21 levels remained undetectable in FGF21KO mice transplanted with WT SAT. This result also supports the indirect effects of subcutaneous fat transplantation. Subcutaneous fat transplantation may exert its metabolic benefits in FGF21KO mice by increasing the release of adiponectin and promoting M2 macrophage polarization and its related cytokines. We have now added the result in the main text to have a comprehensive picture of the model (Line 234-235).

- The experimental basis of the statement that recombinant FGF21 is stable, with no degradation, during 4 weeks in the mini-pump (last Answer) is not provided. Was it checked in the referenced previous paper Lin Z et al?.

Answer: Yes, the stability of rmFGF21 in mini-pump was checked in the referenced previous paper (Lin Z et al). Four weeks after mini-pump was implanted, the mouse was sacrificed and the pump was taken out. Residual rmFGF21 in the pump was checked by western blot. We observed the protein was still intact, without any degradation.

- The involvement of PPARgamma pathway should be toned down relative to the conclusive statements (including the last picture). The panel of transcripts depicted as "PPARgamma-targets" are not full standard and, in any case, a further experimentation beyond this type of very indirect approach (the genes tested have plenty of other transcriptional regulators) is needed for a full-blown claim for PPARgamma role.

Answer: We have toned down the conclusive statement regarding PPAR γ pathway (including Figure 9) in the manuscript.

Responses to Reviewer #3:

The revised manuscript by Li et al. entitled "Fibroblast growth factor 21 increases insulin sensitivity through specific expansion of subcutaneous fat" has been substantially strengthened by several additional experiments suggested by the reviewers.

All of the points made in my initial review have now been appropriately addressed by the authors. First, the authors now provide a clear rationale for the novel conceptual advance made by this paper. They explain that this paper is the first to show that FGF21 at physiological doses (as opposed to pharmacological doses in many studies) appears to

improve insulin sensitivity via hyperplasia of subcutaneous white fat. Although they do not directly measure hyperplasia, which can be done via BrdU incorporation, this is a reasonable inference to make from their morphological characterization and in vitro studies. Second, they have now generated adipocyte-specific beta-klotho knockout mice to show genetically that the benefits of physiological FGF21 are mediated via action on adipocytes. They have also satisfactorily addressed all of the minor points raised in my review.

I also want to point out, in reading the response to the other reviewers, that I think the authors have now shown that the actions of physiological FGF21 are unlikely to be mediated via effects on brown fat or beige fat biogenesis. Their new data includes UCP1 Western blots and immunostaining as shown in supplementary Figure 4. If they wanted to be even more definitive, this data could be further supported by measuring RNA levels of genes involved in the thermogenic pathway in subcutaneous fat and brown fat. Moreover, they could do tissue respiration assays to show that physiological FGF21 does not lead to increased O₂ consumption in subcutaneous fat or brown fat.

Overall, this is now an interesting manuscript that represents a nice advance in molecular metabolism. My only other suggestion would be to have the paper thoroughly edited for grammar and style to make it more straightforward to read.

Answer: Thank you very much for your positive feedback and invaluable suggestions. We agree that BrdU incorporation method directly detects cellular proliferation in adipocytes. We will carry out these experiments in the future to further support our conclusion.

In our study, we show that there is no change of UCP1 levels in SAT or BAT subcutaneous fat and brown fat after replenishing physiological rmFGF21 in FGF21KO mice. We measured mRNA levels of genes involved in the thermogenic pathway (*Cidea*, *Cox7a*, *Cox8b* and *DIO2*) in subcutaneous fat and brown fat. No significant changes were found among the three groups. Additionally, we analyzed the overall energy expenditure and oxygen consumption and no significant differences were found after replenishing physiological rmFGF21, indirectly indicating that physiological dose of rmFGF21 did not lead to increased O₂ consumption. As it will take additional several months to repeat another round of animal experiments to carry out the tissue respiration assays, we prefer to modify our statements in the manuscript (Line 209-213).

We have edited the paper carefully for grammar and style to make it more straightforward to read.

Reviewers' Comments:

Reviewer #2:

Remarks to the Author:

The additions and replies to the comments in my last report in the revised manuscript are appropriate and fulfill my concerns. Moreover, the way that results are presented is enough balanced in light of the results provided.